# Metric from Human: Zero-shot Monocular Metric Depth Estimation via Test-time Adaptation

**Yizhou Zhao**[1], **Hengwei Bian**[1], **Kaihua Chen**[1], **Pengliang Ji**[1], **Liao Qu**[1], **Shao-yu Lin**[1],
**Weichen Yu**[1], **Haoran Li**[2], **Hao Chen**[1], **Jun Shen**[2], **Bhiksha Raj**[1], **Min Xu**[1*]
[1]Carnegie Mellon University, Pittsburgh       [2]University of Wollongong, Wollongong
https://github.com/Skaldak/MfH

## Abstract

Monocular depth estimation (MDE) is fundamental for deriving 3D scene structures from 2D images. While state-of-the-art monocular relative depth estimation (MRDE) excels in estimating relative depths for in-the-wild images, current monocular metric depth estimation (MMDE) approaches still face challenges in handling unseen scenes. Since MMDE can be viewed as the composition of MRDE and metric scale recovery, we attribute this difficulty to scene dependency, where MMDE models rely on scenes observed during supervised training for predicting scene scales during inference. To address this issue, we propose to use humans as landmarks for distilling scene-independent metric scale priors from generative painting models. Our approach, Metric from Human (MfH), bridges from generalizable MRDE to zero-shot MMDE in a generate-and-estimate manner. Specifically, MfH generates humans on the input image with generative painting and estimates human dimensions with an off-the-shelf human mesh recovery (HMR) model. Based on MRDE predictions, it propagates the metric information from painted humans to the contexts, resulting in metric depth estimations for the original input. Through this annotation-free test-time adaptation, MfH achieves superior zero-shot performance in MMDE, demonstrating its strong generalization ability.

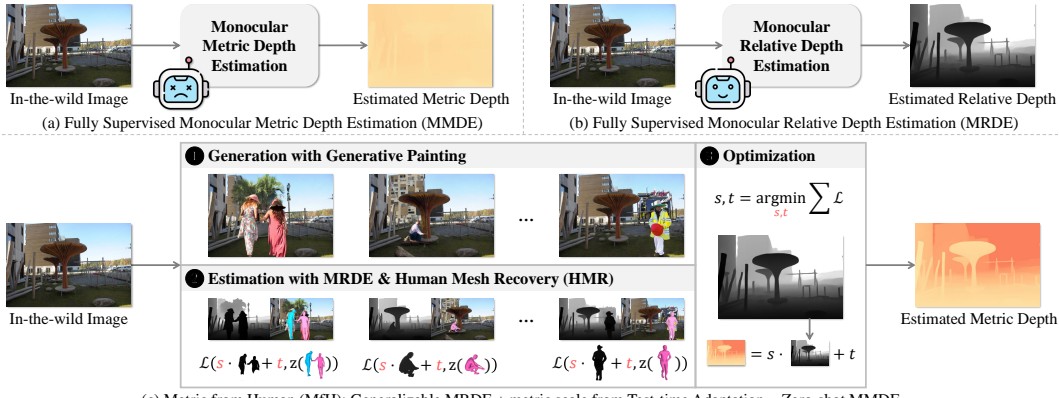

Figure 1: **Illustration of our motivation.** (a) Fully supervised MMDE cannot generalize well on unseen data as (b) MRDE, with its reliance on training scenes for predicting metric scales during test time. (c) Hence, we develop MfH to distill metric scale priors from generative models in a generate-and-estimate manner, bridging the gap from generalizable MRDE to zero-shot MMDE. We use grayscale to represent normalized depths in MRDE predictions, while a colormap mapping metric depth from meters to RGB values in MMDE results. In ❷, $z(\cdot)$ denotes rasterized metric depths.

---

*Corresponding author.

38th Conference on Neural Information Processing Systems (NeurIPS 2024).

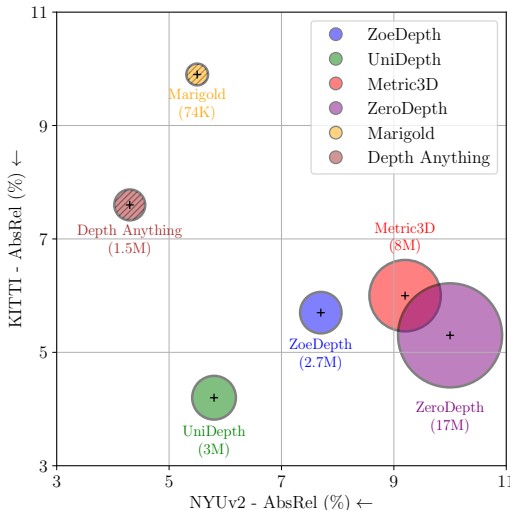

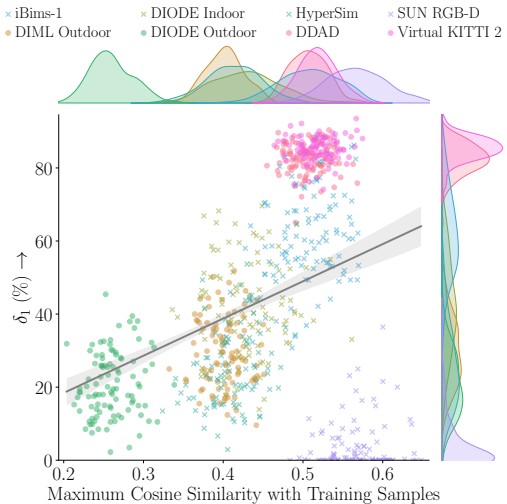

Figure 2: **Comparison of state-of-the-art MRDE and MMDE methods in terms of** AbsRel **and the number of training samples.** Marigold [1] and Depth Anything [2] are designed for MRDE, while the rest are for MMDE. We observe MMDE approaches require notably more data to achieve similar AbsRel as MRDE.

Figure 3: **MMDE $\delta_1$ versus the maximum cosine similarity between each test sample and all metric-annotated training samples.** "$\times/\circ$": from indoor/outdoor datasets. We see that the scale-related performance of a test sample positively correlates with its similarity to training samples. Details can be found in Appendix A.1.

# 1 Introduction

Monocular depth estimation (MDE) is essential in understanding the 3D structure of scenes from 2D images and has many applications in robotics [3, 4], autonomous driving [5, 6], and virtual reality [7, 8]. It requires recovering depth information from a single image without relying on additional sensors or stereo cameras, thereby being inherently ill-posed.

Recent literature mainly explores MDE in two branches, namely, monocular relative depth estimation (MRDE) [2, 1, 9] and monocular metric depth estimation (MMDE) [10–17]. MRDE estimates normalized depths or disparities by factoring out the scale. Its scale-invariant nature enables large-scale training on diverse datasets with distinct camera parameters, while at the cost of bringing in scale ambiguity. In contrast, MMDE predicts absolute depths in *meters*. Due to the unbounded output range and the intertwined relationship between depths and focal lengths, early works of this line often cannot perform well on test data with arbitrary scene scales or camera intrinsics. To compensate for this, recent progress resorts to injecting scene information [12] or camera information [10, 11, 14] into the model. The former attempt learns scene-specific scale priors, modeled with metric heads for indoor or outdoor scenes, and uses either to transform relative depths into metric depths in a heuristic manner. The latter aims to disambiguate scale prediction with extra camera inputs. However, as shown in Fig. 2, both lines of work require notably larger amounts of labeled training data to achieve similar mean absolute relative errors (AbsRel) as their MRDE counterparts.

*What causes the data hunger of MMDE*, and *what makes MMDE harder to generalize*? Given that MMDE can be viewed as the composition of MRDE and metric scale recovery, we posit the latter might be the primary factor. MMDE models might face challenges in inferring scene scales without sufficient exposure to similar annotated scenes during training, which is not a problem for scale-invariant MRDE. To validate our assumption, we evaluate a scale-related metric ($\delta_1$) of an MMDE model, ZoeDepth [12], on randomly sampled test images. Meanwhile, we calculate the maximum cosine similarity between each test sample and all training samples with metric annotations using DINOv2 [18]. Our findings from Fig. 3 indicate a clear trend: higher similarity to training samples positively correlates with better performance, and vice versa. This reflects a scene dependency of MMDE models, likely arising from their supervised training paradigm. In other words, they tend to learn an implicit mapping between training scenes and metric scales from <image, metric annotation> pairs. As a result, adapting to novel scenes may require extra domain-specific fine-tuning.

To address this dependency for better generalization capabilities, we propose to avoid scene-dependent supervised learning, while leveraging a scene-independent metric scale prior. Our insights are two-fold. First, we observe generative painting models can paint objects of proper sizes based on partial contexts, indicating an underlying sense of scales. Additionally, humans can be potentially utilized as relatively universal landmarks, since humans exhibit sizes that are generally more comparable to each other than other common in-the-wild objects, e.g., tables, trees, and cars. To explicitly derive a metric scale prior from generative painting models, we notice state-of-the-art human mesh recovery (HMR) approaches [19–21] can robustly estimate human dimensions for in-the-wild images. Also, they typically output SMPL [22, 23] representations with shape space defined in *meters*. While the input image does not guarantee to include humans, we speculate an off-the-shelf image painting model can paint proportionate humans in the scene, which provides an opportunity to retrieve metric-scale information for the original input by measuring painted humans. Hence, we introduce a test-time adaptation pipeline, Metric from Human (MfH), as illustrated in Fig. 1. Concretely, it 1) paints humans with partial contexts of the input image, 2) estimates human dimensions from the painted image, and 3) propagates the metric-scale information from humans to the contexts for MMDE. In this way, we can distill the metric scale prior hidden inside the generative painting model, unleashing its power to comprehend diverse scenes. As a result, our MfH mitigates the scene dependency issue in fully supervised MMDE, thereby being potentially more generalizable to unseen scenes.

Our contributions can be summed up as follows:

1. We discuss that the current obstacle for generalizable MMDE lies in scene dependency and propose to use a scene-independent metric scale prior as a solution. Further, we find it possible to establish such prior by distilling from generative painting models.

2. To extract the metric scale prior from generative painting models for zero-shot MMDE, we design a test-time adaptation framework, Metric from Human (MfH). Using humans as landmarks, we bridge from MRDE to MMDE by a generate-and-estimate pipeline.

3. Through qualitative and quantitative experiments, we demonstrate the superiority and generalization ability of our MfH in zero-shot MMDE, needless of any metric depth annotations.

## 2   Related Work

**Monocular Depth Estimation (MDE)** has garnered significant interest in recent years. Early approaches focused on supervised methods that predict either monocular metric depth estimation (MMDE) [16, 24–27, 15] or monocular relative depth estimation (MRDE) [27–29, 9]. Despite remarkable progress in network architectures [30–33, 16, 13, 34, 15], existing MMDE methods often confine their training and testing to specific domains, leading to performance degradation under minor domain shifts and poor generalization to unseen environments. In contrast, relative depth models have demonstrated better generalization by leveraging scale-invariant losses [9, 35, 36] on diverse datasets. However, these models cannot recover metric scales, which are crucial for downstream applications. Recent works explored generalizable MMDE models [12, 11, 14, 10] for diverse domains, leveraging camera awareness through explicit incorporation of intrinsics [37, 11] or normalization based on camera properties [38, 17, 14]. They often require fine-tuning to adjust to specific domains [11, 10]. Several recent studies explore zero-shot MMDE, using language as a prior to ground predictions to metric scale [39–41]. However, their hand-crafted depth captions to connect the language and metric worlds are often too coarse to capture accurate depths. Our MfH instead distills metric scale priors from generative painting models, enhancing both the generalization capability and pixel-wise precision of zero-shot MMDE models without relying on metric depth annotations.

**Human Mesh Recovery (HMR)** aims to reconstruct 3D human bodies from visual inputs. Optimization-based HMR relies on iterative optimization techniques to fit parametric body models such as SMPL to detect image features. Examples include SMPLify [42] and its variants [23, 43], which iteratively minimize an objective function to align the model with 2D key points and silhouettes. In contrast, feed-forward methods [44–49] directly regress the body shape and pose parameters from a single image using deep learning techniques. Among them, HMR 2.0 [20] is a fully transformer-based approach for recovering 3D human meshes from single images. We adopt it as our HMR model for estimating in-the-wild human structures and poses. With HMR, we derive metric scale priors from generative painting models, thereby bridging generalizable MRDE to zero-shot MMDE.

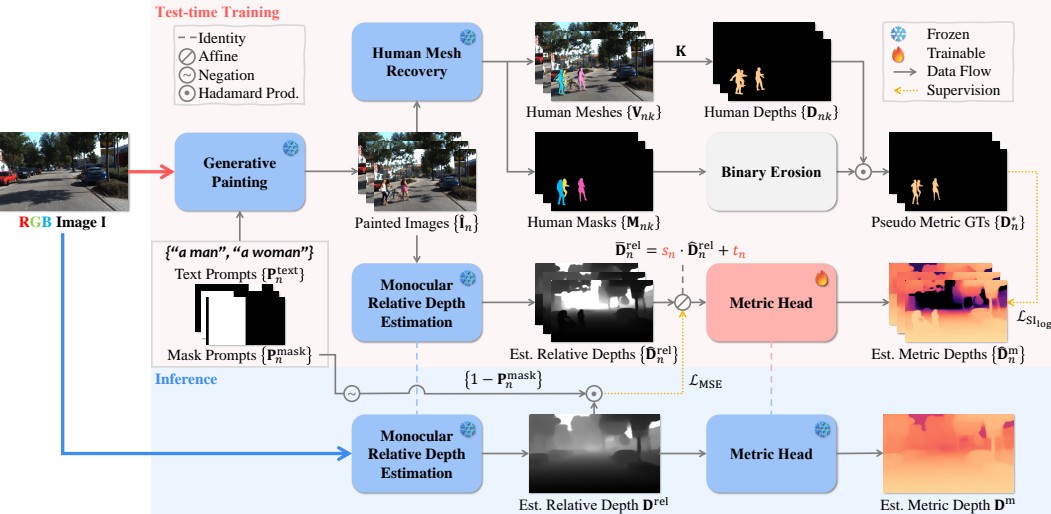

Figure 4: **The framework of Metric from Human (MfH).** Our pipeline comprises two phases. The test-time training phase learns a metric head that transforms relative depths into metric depths based on images randomly painted upon the input image and the corresponding pseudo ground truths. After training the metric head, the inference phase estimates metric depths for the original input.

## 3 Method

Taking an RGB image $\mathbf{I} \in \mathbb{R}^{H \times W \times 3}$ with its camera intrinsic $\mathbf{K} \in \mathbb{R}^{3 \times 3}$ as input, we aim to estimate its pixel-wise metric depths $\mathbf{D}^{\mathrm{m}} \in \mathbb{R}^{H \times W \times 1}$. Unlike existing MMDE methods [12, 11, 14, 10] that typically train on images with metric annotations and expect the trained model to generalize to unseen inputs, we instead consider a test-time adaptation scenario. That is, to estimate metric depths on a certain image without training on domain-specific metric annotations. To achieve this, we propose a framework to learn a metric head for each input $\mathbf{I}$, as depicted in Fig. 4. The metric head is concatenated after an off-the-shelf MRDE model to transform relative depths into metric depths. While the inference pipeline is simplistic, our key insight is to use humans as metric landmarks during test-time training. Since humans are not guaranteed to exist in in-the-wild images, we introduce a generate-and-estimate method in Sec. 3.2 to extract metric scale information from the given image. This extracted information together with the estimated relative depths allows us to approach annotation-free zero-shot metric depth estimation, as outlined in Sec. 3.3.

### 3.1 Preliminaries

#### 3.1.1 Monocular Depth Estimation (MDE)

Assuming a pinhole camera model, we have the following relation

$$\mathbf{D}^{\mathrm{m}} = \frac{b \cdot f}{\mathbf{d}^{\mathrm{m}}}, \ \mathbf{D}^{\mathrm{rel}} = \mathrm{rel}(\mathbf{D}^{\mathrm{m}}) = \frac{\mathbf{D}^{\mathrm{m}} - \mathrm{t}(\mathbf{D}^{\mathrm{m}})}{\mathrm{s}(\mathbf{D}^{\mathrm{m}})}, \tag{1}$$

where $b$ and $f$ are the camera baseline and focal length, $\mathbf{D}^{\mathrm{m}}$ and $\mathbf{d}^{\mathrm{m}}$ are metric depths and disparities, and $\mathrm{s}(\cdot)$ and $\mathrm{t}(\cdot)$ are scalar functions, denoting a scale and a translation to normalize the input. We use superscript $^{\mathrm{m}}$ and $^{\mathrm{rel}}$ to refer to metric and relative values, accordingly. Due to the correlation between camera parameters and the scale of depth, MMDE cannot generalize well if the camera intrinsic is unknown or the scene scale is hard to predict, e.g., when the scene is unseen during training. In contrast, MRDE enjoys better generalization ability for its affine-invariant formulation.

#### 3.1.2 Human Mesh Recovery (HMR)

We adopt a state-of-the-art HMR model, HMR 2.0 [20], for reconstructing camera-frame human meshes from an image $\hat{\mathbf{I}}_n \in \mathbb{R}^{H \times W \times 3}$ with $K$ people. Starting with human segmentation masks $\{\mathbf{M}_{nk} \in \mathbb{R}^{H \times W \times 1}\}_{k=1}^{K}$ from Mask R-CNN [50], HMR 2.0 predicts SMPL [22] parameters for each

human as $\{\boldsymbol{\Phi}_{nk}, \boldsymbol{\theta}_{nk}, \boldsymbol{\beta}_{nk}, \boldsymbol{\Gamma}_{nk}\}_{k=1}^{K}$. These parameters represent global orientation $\boldsymbol{\Phi}_{nk} \in \mathbb{R}^{3\times3}$, body pose $\boldsymbol{\theta}_{nk} \in \mathbb{R}^{22\times3\times3}$, shape $\boldsymbol{\beta}_{nk} \in \mathbb{R}^{10}$, and root translation $\boldsymbol{\Gamma}_{nk} \in \mathbb{R}^{3}$. Then the human body meshes with vertices $\mathbf{V}_{nk} \in \mathbb{R}^{3\times6890}$ can be recovered with the SMPL model

$$\mathbf{V}_{nk} = \text{SMPL}\left(\boldsymbol{\Phi}_{nk}, \boldsymbol{\theta}_{nk}, \boldsymbol{\beta}_{nk}\right) + \boldsymbol{\Gamma}_{nk}, \text{ where } k = 1, 2, \ldots, K. \tag{2}$$

Since the shape space of SMPL is defined in *meters*, the generated vertices $\{\mathbf{V}_{nk}\}$ are also in *meters*. As a result, all HMR regressors, such as HMR 2.0 we use, inherit such data prior.

## 3.2 Generating Humans as Metric Landmarks

To start off, we randomly place people on the input image with generative image painting, with text prompts $\{\mathbf{P}_n^{\text{text}}\}_{n=1}^{N}$ sampled from {"a man", "a woman"} and mask prompts $\{\mathbf{P}_n^{\text{mask}} \in \mathbb{R}^{H\times W\times1}\}_{n=1}^{N}$ sampled from rectangles smaller than the whole image. We write $\mathbf{P}_n = (\mathbf{P}_n^{\text{text}}, \mathbf{P}_n^{\text{mask}})$ as a shorthand. Once at a time, we randomly generate $N$ painted images with humans

$$\hat{\mathbf{I}}_n = \text{paint}\left(\mathbf{I}|\mathbf{P}_n\right), \text{ where } n = 1, 2, \ldots, N, \tag{3}$$

and $\{\hat{\mathbf{I}}_n \in \mathbb{R}^{H\times W\times3}\}_{n=1}^{N}$ are the painted images. We observe the generative image painting model can paint people of proper sizes that are compatible with the unmasked background. This allows us to use the painted people as landmarks to inform our model of the metric scale. To this end, we fed these painted images into HMR 2.0 to predict human instance segmentation masks $\{\mathbf{M}_{nk}\}_{n=1,k=1}^{N,K}$ and meshes $\{\mathbf{V}_{nk}\}_{n=1,k=1}^{N,K}$, where the subscripts denote the $k$-th person in the $n$-th image. We obtain the pseudo metric ground truths $\mathbf{D}_n^*$ with rasterization

$$\mathbf{D}_n^* = \min_k \left[\text{erode}\left(\mathbf{M}_{nk} \cap \mathbf{S}_{nk}\right) \odot \mathbf{D}_{nk}\right], \tag{4}$$

where $\mathbf{S}_{nk}, \mathbf{D}_{nk} = \rho(\mathbf{K}, \mathbf{V}_{nk})$ are the rasterized silhouettes and depths, respectively, and $\odot$ stands for the Hadamard product. We erode the intersection of the instance segmentation mask $\mathbf{M}_{nk}$ and the rasterized silhouette $\mathbf{S}_{nk}$ to avoid overlapping and take the minimum of $k$ depth maps so that the depth values from closer humans can occlude the farther ones. Using these pseudo ground truths, we supervise the learning of the metric head with the scale-invariant log ($\text{SI}_{\text{log}}$) loss [51]

$$\mathcal{L}_{\text{SI}_{\text{log}}}(\hat{\mathbf{D}}_n^{\text{m}}, \mathbf{D}_n^*) = \frac{1}{HW} \sum_i \epsilon_i^2 - \frac{\lambda}{(HW)^2} \left(\sum_i \epsilon_i\right)^2, \tag{5}$$

where $\epsilon = \log \hat{\mathbf{D}}_n^{\text{m}} - \log \mathbf{D}_n^*$, with $\hat{\mathbf{D}}_n^{\text{m}}$ being the estimated metric depth map, subscript $_i$ denotes the index of each pixel, and $\lambda \in [0, 1]$. Since the rasterized depths $\mathbf{D}_{nk}$ are in *meters* and the first term in $\mathcal{L}_{\text{SI}_{\text{log}}}$ is pixel-wise $l_2$, this loss provides crucial metric scale information to the metric head.

## 3.3 Transforming Relative Depths into Metric Depths

During training, we estimate a relative depth map $\hat{\mathbf{D}}_n^{\text{rel}}$ for each painted image $\hat{\mathbf{I}}_n$ with a pre-trained MRDE model, and learn a metric head to transform $\hat{\mathbf{D}}_n^{\text{rel}}$ into an estimated metric depth map $\hat{\mathbf{D}}_n^{\text{m}}$. Similarly, we obtain $\mathbf{D}^{\text{rel}}$ and $\mathbf{D}^{\text{m}}$ from the original input $\mathbf{I}$ during inference. According to Eq. (1), we can learn a simple linear layer as our metric head, i.e., $\hat{\mathbf{D}}_n^{\text{m}} = \hat{s} \cdot \hat{\mathbf{D}}_n^{\text{rel}} + \hat{t}$. In addition, we need to account for the difference between the relative depths predicted from the original image $\mathbf{D}^{\text{rel}}$ and those from the painted images $\{\hat{\mathbf{D}}_n^{\text{rel}}\}$ on the unpainted regions. Considering the affine-invariant nature of MRDE, we further decompose metric depth predictions with

$$\hat{\mathbf{D}}_n^{\text{m}} = s \cdot (s_n \cdot \hat{\mathbf{D}}_n^{\text{rel}} + t_n) + t = s \cdot \bar{\mathbf{D}}_n^{\text{rel}} + t, \text{ where } \bar{\mathbf{D}}_n^{\text{rel}} = s_n \cdot \hat{\mathbf{D}}_n^{\text{rel}} + t_n, \tag{6}$$

and $\{s_n\}, \{t_n\}, s, t \in \mathbb{R}$ are optimizable parameters. Then we align $\mathbf{D}^{\text{rel}}$ and $\{\bar{\mathbf{D}}_n^{\text{rel}}\}$ with

$$\mathcal{L}_{\text{MSE}}(\bar{\mathbf{D}}_n^{\text{rel}}, \mathbf{D}^{\text{rel}}) = \left\|\left(1 - \mathbf{P}_n^{\text{mask}}\right) \odot \left(\bar{\mathbf{D}}_n^{\text{rel}} - \mathbf{D}^{\text{rel}}\right)\right\|_2. \tag{7}$$

This objective considers the pixel-wise alignment of unpainted regions with an affine transform specific to each painted image. Finally, we formulate our complete objective function as

$$\min_{s,t} \sum_n \mathcal{L}_{\text{SI}_{\text{log}}}(\hat{\mathbf{D}}_n^{\text{m}}, \mathbf{D}_n^*) + \sum_n \min_{s_n,t_n} \mathcal{L}_{\text{MSE}}(\bar{\mathbf{D}}_n^{\text{rel}}, \mathbf{D}^{\text{rel}}). \tag{8}$$

Table 1: Performance comparisons of our MfH and state-of-the-art methods on the NYU-Depth V2 [52] and KITTI [55] datasets. †LORN uses 200 images and 2,500 partial images for training.

| Method | Supervision | NYUv2 | | | | KITTI | | | |
|---|---|---|---|---|---|---|---|---|---|
| | | $\delta_1 \uparrow$ | AbsRel $\downarrow$ | $SI_{log} \downarrow$ | RMSE $\downarrow$ | $\delta_1 \uparrow$ | AbsRel $\downarrow$ | $SI_{log} \downarrow$ | RMSE $\downarrow$ |
| ZeroDepth [11] | many-shot | 90.1 | 10.0 | — | 0.380 | 89.2 | 10.2 | — | 4.38 |
| Metric3D [14] | many-shot | 92.6 | 9.38 | 9.13 | 0.337 | 97.5 | 5.33 | 7.28 | 2.26 |
| UniDepth-C [10] | many-shot | 97.2 | 6.26 | 6.41 | 0.232 | 97.9 | 4.69 | 6.71 | 2.00 |
| UniDepth-V [10] | many-shot | 98.4 | 5.78 | 5.27 | 0.201 | 98.6 | 4.21 | 5.84 | 1.75 |
| LORN [58]† | few-shot | 70.3 | 101 | — | 9.452 | — | — | — | — |
| Hu et al. [40] | one-shot | 42.8 | 34.7 | — | 1.049 | 31.2 | 38.4 | — | 12.29 |
| DepthCLIP [59] | zero-shot | 39.4 | 38.8 | — | 1.167 | 28.1 | 47.3 | — | 12.96 |
| MfH (Ours) | zero-shot | **83.2** | **13.7** | **9.78** | **0.487** | **81.2** | **13.3** | **10.5** | **4.21** |

Such formulation propagates metric scale information from human pixels to background pixels, thereby enabling the metric head to predict for the non-human context. After optimization, it is then possible to infer metric depths for the original input image. While one can deploy fancier metric head and apply up-to-affine consistency constraints between aligned relative depth predictions $\bar{D}_n^{rel}$ and metric depth predictions $\hat{D}_n^m$ for better robustness, we observe a simple affine transform is capable of providing good predictions, leaving further parameterization for future works.

## 4 Experiments

### 4.1 Experimental Setting

**Datasets.** Under our test-time adaptation setting, we do not train on any datasets but only test on each input image directly after image-specific optimizations. Specifically, we evaluate the zero-shot MMDE capability of MfH on NYU-Depth V2 [52], IBims-1 [53], ETH-3D [54] with the split from [13] and official masks, and KITTI [55] with the corrected Eigen-split from [51]. Following prior works [12, 15, 16], we apply the Eigen evaluation mask [51] on NYU-Depth V2 and IBims-1 while the Garg evaluation mask [56] on KITTI.

**Evaluation Metrics.** We employ several common metrics to assess the performance of all baseline methods and our model. The $\delta_1 = \frac{1}{HW} \sum_{i=1}^{HW} \left[ \max \left( \frac{D_{pred}}{D_{gt}}, \frac{D_{gt}}{D_{pred}} \right) < 1.25 \right]$ metric evaluates the fraction of predicted depth values that are within a threshold factor of their corresponding true values; the Mean Absolute Relative Error, AbsRel $= \frac{1}{HW} \sum_{i=1}^{HW} \frac{|D_{gt} - D_{pred}|}{D_{gt}}$, measures the average absolute difference between the predicted and true depth values, normalized by the true depth; the Scale Invariant Logarithmic Error, $SI_{log} = 100\sqrt{Var(\log D_{pred} - \log D_{gt})}$, quantifies the error in a logarithmic scale that is invariant to the absolute scale of the scene; the Root Mean Squared Error, RMSE $= \sqrt{\frac{1}{HW} \sum_{i=1}^{HW} (D_{gt} - D_{pred})^2}$, focuses on the square root of the mean of squared differences between the predicted and actual depth values, emphasizing larger errors.

**Implementation Details.** We adopt Depth Anything [2] without fintuning on metric annotations as our MRDE model, Stable Diffusion v2 [57] for generative painting, and HMR 2.0 [20] for human mesh recovery. In $\mathcal{L}_{SI_{log}}$, we follow ZoeDepth [12] to set the $\lambda = 0.15$. For optimizing the alignment parameters $\{s_n\}, \{t_n\}$, we leverage linear regression to obtain a close-formed solution. As for optimizing the metric head parameters, $s, t$, we use the L-BFGS optimizer with a fixed learning rate of 1 for 50 steps. Unless otherwise specified, we randomly paint 32 images for our comparison experiments and 4 for our ablation studies. All experiments are run on one NVIDIA A100 GPU.

### 4.2 Comparison Results

We first evaluate the MMDE results on NYU-Depth V2 [52] and KITTI [55], common benchmarks with indoor and outdoor scenes, respectively. In Tab. 1, we show that our zero-shot MfH consistently outperforms other approaches trained with few/one/zero-shot supervision. This indicates that generative painting models are capable of capturing metric scale information, which is potentially more accurate than that embedded in language models [40, 59]. With our generate-and-estimate pipeline, the metric scale prior hidden inside generative painting models can be leveraged for zero-shot MMDE.

Table 2: Performance comparisons of our MfH and many-shot methods on the DIODE (Indoor) [60], iBims-1 [61], and ETH3D [54] datasets. *-{N, K, NK}: fine-tuned on NYUv2 [52], KITTI [55], or the union of them. We re-evaluate all results with a consistent pipeline for metric completeness.

| Method | DIODE (Indoor) | | | | iBims-1 | | | | ETH3D | | | |
|---|---|---|---|---|---|---|---|---|---|---|---|---|
| | $\delta_1 \uparrow$ | AbsRel ↓ | $SI_{log} \downarrow$ | RMSE ↓ | $\delta_1 \uparrow$ | AbsRel ↓ | $SI_{log} \downarrow$ | RMSE ↓ | $\delta_1 \uparrow$ | AbsRel ↓ | $SI_{log} \downarrow$ | RMSE ↓ |
| ZoeDepth-NK [12] | 38.8 | 33.0 | 13.3 | 1.598 | 61.0 | 18.7 | 8.98 | 0.778 | 33.5 | 47.3 | 14.0 | 2.094 |
| Depth Anything-N [2] | 29.7 | 32.7 | 12.5 | 1.486 | 71.3 | 15.0 | 7.58 | 0.594 | 25.2 | 38.7 | 10.2 | 2.327 |
| Depth Anything-K [2] | 11.1 | 231 | 15.5 | 5.199 | 2.88 | 217 | 17.2 | 5.385 | 16.9 | 136 | 17.1 | 4.202 |
| ZeroDepth [11] | 43.2 | 30.0 | 13.2 | 1.392 | 74.6 | 16.4 | 10.6 | 0.634 | 31.2 | 32.6 | 13.4 | 1.926 |
| Metric3D [14] | − | 26.8 | − | 1.429 | − | **14.4** | − | 0.646 | − | 34.2 | − | 2.965 |
| UniDepth-C [10] | 62.8 | 23.8 | 11.5 | 0.968 | **81.1** | 14.8 | 8.30 | **0.536** | 43.3 | 35.5 | 10.3 | 1.532 |
| UniDepth-V [10] | **79.8** | **18.1** | **10.4** | **0.760** | 23.4 | 35.7 | **6.87** | 1.063 | 27.2 | 43.1 | 8.93 | 1.950 |
| MfH (Ours) | 42.2 | 34.5 | 13.2 | 1.363 | 67.7 | 23.3 | 9.73 | 0.738 | **47.1** | **24.0** | **8.16** | **1.366** |

Table 3: Ablation study for MRDE models and optimization targets on the NYUv2 dataset. True depth/disparity represents the performance with oracle depths/disparities as optimization targets.

| MRDE Model | Optim. Target | $\delta_1 \uparrow$ | AbsRel ↓ | $SI_{log} \downarrow$ | RMSE ↓ |
|---|---|---|---|---|---|
| ZoeDepth [12] | true depth | 63.9 | 22.4 | 24.5 | 0.692 |
| | true disparity | 66.5 | 20.1 | 23.2 | 0.658 |
| | painted depth | 29.5 | 35.5 | 30.6 | 1.246 |
| | painted disparity | 30.2 | 32.3 | 25.1 | 1.158 |
| Marigold [1] | true depth | 96.3 | 5.88 | 8.70 | 0.251 |
| | true disparity | 78.8 | 15.1 | 20.9 | 0.523 |
| | painted depth | 60.7 | 21.4 | 19.5 | 0.689 |
| | painted disparity | 40.7 | 31.1 | 31.3 | 1.086 |
| Depth Anything [2] | true depth | 75.8 | 16.8 | 19.8 | 0.657 |
| | true disparity | 97.9 | 4.57 | 6.68 | 0.222 |
| | painted depth | 50.1 | 41.7 | 27.9 | 1.874 |
| | painted disparity | 66.8 | 21.9 | 15.2 | 0.792 |

Table 4: Ablation study for optimization parameters and optimization targets on NYUv2. We optimize the predictions in the same space as optimization targets, i.e., the depth space for depth targets and the inverted depth space for disparity targets. The same applies to Tab. 3

| Optim. Param. | Optim. Target | $\delta_1 \uparrow$ | AbsRel ↓ | $SI_{log} \downarrow$ | RMSE ↓ |
|---|---|---|---|---|---|
| $s, t$ | true disparity | 97.9 | 4.57 | 6.68 | 0.222 |
| | painted disparity | 58.6 | 28.0 | 21.2 | 1.079 |
| $\{s_n\}, \{t_n\}, s$ | true disparity | 29.2 | 43.5 | 47.3 | 1.910 |
| | painted disparity | 29.6 | 74.4 | 36.6 | 3.000 |
| $\{s_n\}, \{t_n\}, t$ | true disparity | 0.00 | 98.9 | 62.5 | 2.825 |
| | painted disparity | 0.30 | 113 | 182 | 3.387 |
| $\{s_n\}, \{t_n\}, s, t$ | true disparity | 97.9 | 4.57 | 6.68 | 0.222 |
| | painted disparity | 66.8 | 21.9 | 15.2 | 0.792 |

In Tab. 2, we further provide a comparison between MfH and state-of-the-art many-shot methods, which are typically trained upon large-scale datasets with dense metric depth annotations. Our model achieves performance on par with, and sometimes superior to, these approaches, especially on ETH3D [54], which contains both indoor and outdoor scenes. It is noteworthy that these prior arts can do well on certain datasets while failing on others. For instance, ZeroDepth [11] performs well on iBims-1 but struggles to estimate depths on DIODE (Indoor) and ETH3D accurately. Similarly, UniDepth-V [10] shows promising results on DIODE (Indoor) while underperforming on the other two benchmarks. These findings signify the scene-dependent nature of existing fully supervised methods, which may result in degraded performance on unseen scenes. In contrast, our model demonstrates robust zero-shot generalization capabilities across diverse scenes. We further highlight the comparison among our MfH and Depth Anything [2] fine-tuned on NYUv2 or KITTI (2nd-3rd rows). These methods adopt a common Depth Anything MRDE backbone while deploying different strategies for MMDE. The results demonstrate that our test-time adaptation strategy generally works better than domain-specific fine-tuning, without the need for training on metric depth annotations.

## 4.3  Ablation Study

**Impact of MRDE models.** In Tab. 3, we investigate the performance of MfH with different MRDE models and various optimization targets. For ZoeDepth [12] and Depth Anything [2], we only adopt their pre-trained MRDE backbone as our MRDE model. Note that we use ground truth depths or disparities as optimization targets to show an approximate upper bound of performances, while only accessing painted depths or disparities as pseudo ground truths during real test-time adaptation. We also ensure consistency by optimizing predictions in the same space as the target, i.e., the depth space for depth targets and the inverted depth space for disparity targets. Overall, we observe using Depth Anything as our MRDE model and painted disparities as our optimization target shows the best performance. Comparing the first two rows and the last two rows for each MRDE model, we see that optimizations in the disparity space yield superior results for ZoeDepth [12] and Depth Anything [2], whereas optimizations in the depth space prove more effective for Marigold [1]. A similar scenario can also be found in the last two rows of each MRDE model. This variation in performance may stem from the difference in their output spaces. To be concrete, ZoeDepth and Depth Anything produce inverted depths, while Marigold outputs depths. Optimizing in the original output space can provide better numerical stability, leading to better optimization results.

Table 5: Ablation study for loss functions on NYUv2.

Figure 5: **Ablation study for the number of painted images on NYUv2.** Horizontal axis: $N \in \{4, 8, 16, 32\}$ inpainted images.

| Loss | $\delta_1 \uparrow$ | AbsRel $\downarrow$ | $\mathrm{SI}_{\log} \downarrow$ | RMSE $\downarrow$ |
|------|------|--------|---------|-------|
| $l_1$ | 24.2 | 201 | 28.9 | 4.645 |
| $l_2$ | 62.3 | 27.1 | 16.7 | 0.975 |
| $\mathrm{MSE}_{\log}$ | **67.5** | 22.5 | **14.6** | 0.810 |
| $\mathrm{SI}_{\log}$ | 66.8 | **21.9** | 15.2 | **0.792** |

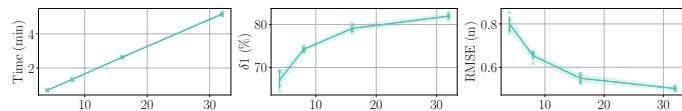

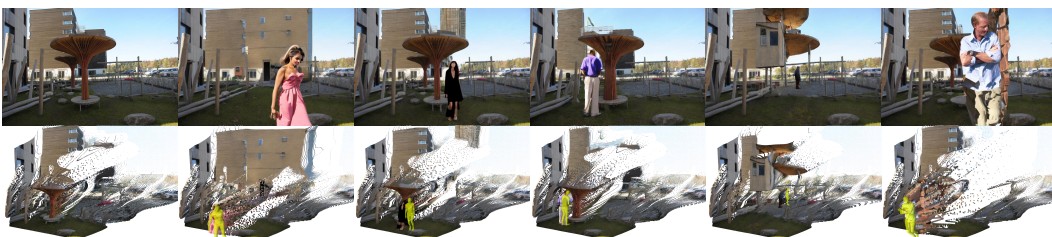

(a) Input & Output      (b) Painted Images & Recovered Human Meshes with Aligned Point Clouds

Figure 6: **An alternative perspective on MfH.** (a) Using camera intrinsics, the estimated metric depths can be transformed into a metric point cloud. (b) Our generate-and-estimate process can be seen as aligning the estimated point cloud of each painted human to its corresponding human mesh.

**Impact of optimization parameters.** In Tab. 4, we verify the benefits of aligning relative depth estimations of the original input and painted images with $\{s_n\}, \{t_n\}$ and parameterizing the metric head with $s, t$. We view the results from bottom to top. First, optimizing all parameters (7th-8th rows) yields the lowest error. When we fix the scaling factor to 1 while keeping the other parameters optimizable (5th-6th rows), the model has the highest error and cannot be aligned with the ground truth. Instead, when using optimizable scales (3rd-4th rows) in the metric head, the model can better capture depths, which indicates an accurate scene scale is crucial in MMDE. Removing the alignment between the input image MRDE and painted image MRDEs (1st-2nd rows) results in sub-optimal predictions since the same contents on two different images might result in distinct MRDE predictions. The performance difference of using different optimization parameters while the true disparity as the target (3rd vs. 5th vs. 7th row) shows it is possible to apply an affine transformation upon MRDE to achieve good MMDE predictions, if with accurate scale and translation recovered.

**Impact of loss functions.** In Tab. 5, we ablate the effect of using various loss functions for test-time training the metric head. Notably, employing an $l_1$ loss (1st row) yields inferior performance compared to losses incorporating an $l_2$ term (2nd-4th rows). This is probably because our generate-and-estimate process can introduce noises to a certain degree. Since the $l_1$ loss treats all errors equally, regardless of their magnitude, it can be more sensitive to small perturbations. An $l_2$ term that focuses more on large errors thus provides better robustness. Furthermore, a comparison between the 2nd and the last two rows shows optimizing in the log space brings better performance, which is expected since logarithmic transformation tends to mitigate the impact of outliers. This also accords with the general experience in training depth models using ground truth annotations, suggesting that the depths of generated humans might also be normally distributed in the log space, akin to real-world scenarios. From the last two rows, we see optimizing with the $\mathrm{MSE}_{\log}$ loss or the $\mathrm{SI}_{\log}$ loss is not discrepant by much. We opt for the $\mathrm{SI}_{\log}$ loss in our optimization process due to its relatively superior $\mathrm{AbsRel}$.

**Impact of painting numbers $N$.** In Fig. 5, we analyze the effect of increasing the number of painted images with humans. Specifically, we paint 4, 8, 16, and 32 images with humans, plotting curves as well as error bars for various metrics against the painting numbers. To draw reliable conclusions, we conduct experiments across five consistent random seeds for each painted image quantity. Our analysis reveals a clear linear association between the per-sample runtime and the number of painted images, as depicted in the time plot. The $\delta_1$ and $\mathrm{RMSE}$ plots show an upward trend in MMDE performance with the increment in painting numbers, albeit sublinearly. By further examining the error bars, we see a larger number of painted images results in better robustness of predictions, which is demonstrated by a smaller, gradually converging standard deviation.

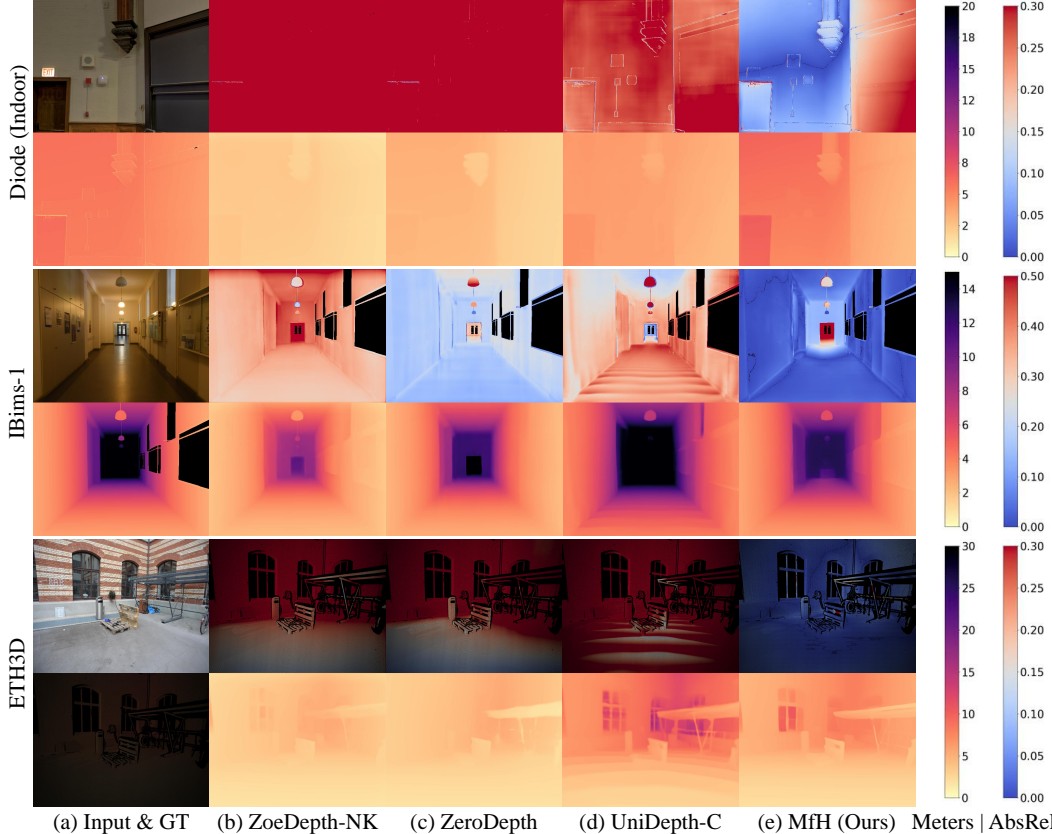

(a) Input & GT   (b) ZoeDepth-NK   (c) ZeroDepth   (d) UniDepth-C   (e) MfH (Ours)   Meters | AbsRel

Figure 7: **Zero-shot qualitative results.** Each pair of consecutive rows corresponds to one test sample. Each odd row shows an input RGB image alongside the absolute relative error map, while each even row shows the ground truth metric depth and predicted metric depths.

## 4.4 Qualitative Analysis

We present an alternative view of our MfH in Fig. 6. With camera intrinsics, the estimated metric depths for both the painted images $\{\hat{\mathbf{D}}_n^{\mathrm{m}}\}$ and the original input $\mathbf{D}^{\mathrm{m}}$ correspond to metric point clouds. Our MfH stretches the point clouds for $\{\hat{\mathbf{D}}_n^{\mathrm{m}}\}$ along the z-axis by setting human landmarks in 3D, revealing the 3D structure of the unpainted background in *meters*. With random painting, we progressively capture the 3D structure of the entire original input, thus bridging MRDE to MMDE.

We demonstrate metric depth predictions and pixel-wise AbsRel in Fig. 7, highlighting the strong zero-shot generalization ability of our MfH. Additionally, we show MMDE results for in-the-wild samples captured by DSLR cameras and smartphones in Fig. 8. Besides the robust performance of MfH, we observe that fully supervised MMDE methods like UniDepth [10] often provide bounded metric depths, inheriting from the limited range of sensors used in their training ground truths. In contrast, our MfH can provide more flexible results.

Case studies, user studies, and more qualitative analysis can be found in Appendix D.

## 5 Conclusion

We present MfH, a method that infers metric depths from in-the-wild images without the need for training on metric depth annotations. Utilizing humans as landmarks to extract metric scale priors from generative painting models, our approach addresses the challenge of scene dependency inherent in MMDE trained with metric depth supervision. Through a test-time adaptation pipeline, MfH effectively captures metric scale information from images by generating and estimating humans, which is then leveraged for zero-shot MMDE. Our experiments demonstrate that MfH achieves superior performance and better generalization ability compared to existing methods.

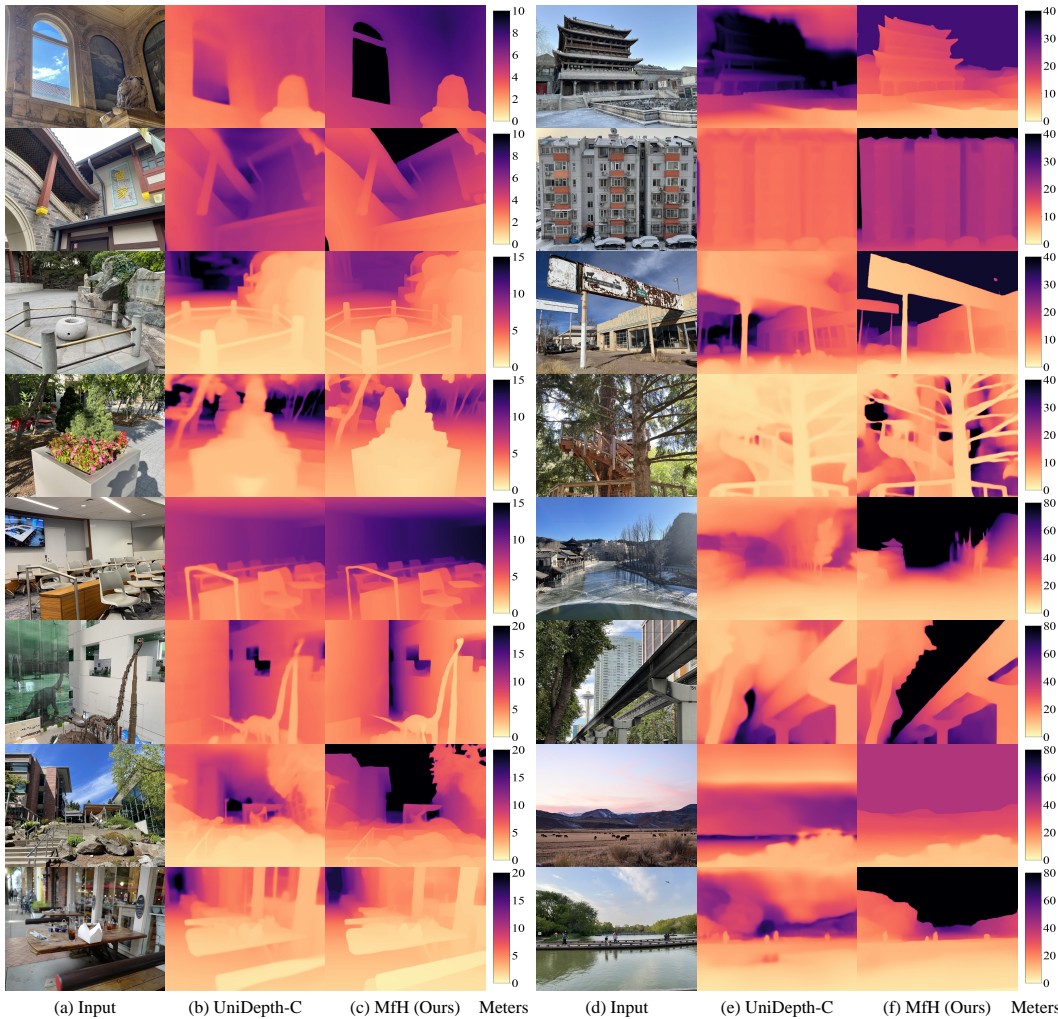

<table>
<tr><td>(a) Input</td><td>(b) UniDepth-C</td><td>(c) MfH (Ours)</td><td>Meters</td><td>(d) Input</td><td>(e) UniDepth-C</td><td>(f) MfH (Ours)</td><td>Meters</td></tr>
</table>

Figure 8: **In-the-wild qualitative results.** Each group of rows (a)-(c) or (d)-(f) corresponds to one in-the-wild test sample captured by a DSLR camera or a smartphone.

**Limitation discussion.** Our MfH works based on the assumption that humans can exist in the scene so that it is possible to paint a human upon the input image. While this holds for most usages of MMDE, it might not be ideal for some cases, e.g., close-up scenes. This opens up new challenges, such as incorporating objects other than humans into the generate-and-estimate pipeline as metric landmarks. Another assumption is the MRDE predictions align with true depths up to affine. Despite the training objectives of MRDE being linearly transformed true depths, the MRDE predictions can contain noises, making the linear metric head hard to capture accurate metric depths. Whether other parameterizations of the metric head can tackle this remains an open question.

**Broader impacts.** Our MfH decreases the demand for metric depth annotation which commonly requires depth sensors or stereo systems, making MMDE models more environmentally friendly. However, its usage of human-related models can perpetuate biases present in the training data, leading to unfair or discriminatory outcomes.

## Acknowledgments and Disclosure of Funding

This study was partially funded by U.S. NIH grants R01GM134020 and P41GM103712, NSF grants DBI-1949629, DBI-2238093, IIS-2007595, IIS-2211597, and MCB-2205148. Additionally, it received support from Oracle Cloud credits and resources provided by Oracle for Research, as well as computational resources from the AMD HPC Fund.

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

# A  Experiments on Scene Dependency

## A.1  Similarity with Training Samples

We provide more details of plotting MMDE $\delta_1$ against the maximum cosine similarity between each test sample and all training samples in Fig. 3. Specifically, we run ZoeDepth [12] on each sample of its test sets and calculate their $\delta_1$ metrics. We also calculate the cosine similarity of each test sample against each training sample with metric annotations with DINOv2 [18] and find the similarity value for the nearest neighbor. Since ZoeDepth first pre-trains its backbone for MRDE and then finetunes on NYUv2 [52] and KITTI [55] for MMDE, we only include the two datasets used for MMDE training during similarity calculation. For the scatter plot, we randomly take 10 different samples each time and plot their average $\delta_1$ and maximum cosine similarity with training samples as one point on the figure. In this way, we plot 100 points for each testing dataset.

## A.2  MRDE Performances of MMDE Models

Table 6: Performance comparisons of our MMDE methods in terms of MMDE and MRDE settings on the DIODE (Indoor) [60], iBims-1 [61], and ETH3D [54] datasets. *-{N, K, NK}: fine-tuned on NYUv2 [52], KITTI [55], or the union of them. We re-evaluate all results with a fair and consistent pipeline for metric completeness.

| Method | Setting | DIODE (Indoor) | | | iBims-1 | | | ETH3D | | |
|---|---|---|---|---|---|---|---|---|---|---|
| | | $\delta_1 \uparrow$ | AbsRel $\downarrow$ | $SI_{log} \downarrow$ | $\delta_1 \uparrow$ | AbsRel $\downarrow$ | $SI_{log} \downarrow$ | $\delta_1 \uparrow$ | AbsRel $\downarrow$ | $SI_{log} \downarrow$ |
| ZoeDepth-NK [12] | MMDE | 38.8 | 33.0 | 13.3 | 61.0 | 18.7 | 8.98 | 33.5 | 47.3 | 14.0 |
| Depth Anything-N [2] | | 29.7 | 32.7 | 12.5 | 71.3 | 15.0 | 7.58 | 25.2 | 38.7 | 10.2 |
| Depth Anything-K [2] | | 11.1 | 231 | 15.5 | 2.88 | 217 | 17.2 | 16.9 | 136 | 17.1 |
| ZeroDepth [11] | | 43.2 | 30.0 | 13.2 | 74.6 | 16.4 | 10.6 | 31.2 | **32.6** | 13.4 |
| Metric3D [14] | | – | 26.8 | – | – | **14.4** | – | – | 34.2 | – |
| UniDepth-C [10] | | 62.8 | 23.8 | 11.5 | **81.1** | 14.8 | 8.30 | **43.3** | 35.5 | 10.3 |
| UniDepth-V [10] | | **79.8** | **18.1** | **10.4** | 23.4 | 35.7 | **6.87** | 27.2 | 43.1 | **8.93** |
| ZoeDepth-NK [12] | MRDE | 91.6 | 12.2 | 11.9 | 97.3 | 5.61 | 7.74 | 94.9 | 8.15 | 9.85 |
| Depth Anything-N [2] | | 94.4 | 10.2 | 10.6 | 98.4 | 4.45 | 6.18 | 93.8 | 8.03 | 9.95 |
| Depth Anything-K [2] | | 92.4 | 11.9 | 12.0 | 95.7 | 6.82 | 9.36 | **97.7** | 6.09 | **7.36** |
| ZeroDepth [11] | | 91.8 | 12.1 | 12.1 | 94.8 | 6.51 | 9.25 | 87.6 | 11.6 | 12.1 |
| UniDepth-C [10] | | 94.2 | 10.2 | 10.7 | 97.6 | 4.68 | 7.24 | 96.6 | 6.96 | 8.76 |
| UniDepth-V [10] | | **95.6** | **8.89** | **9.78** | **98.6** | **3.41** | **5.69** | 97.4 | **5.79** | 7.53 |

To use an MMDE model for MRDE, we align the predictions with ground truths by solving optimal scales and translations. As demonstrated in Tab. 6, under the MRDE setting, MMDE models all perform well on in-the-wild data. In contrast, under the MMDE setting, they degrade to various degrees. These results show that, with fully supervised training, depth models are generally more generalizable under the MRDE setting than the MMDE setting. Since the MMDE task can be understood as a combination of MRDE and metric scale recovery, we believe the latter can be more difficult to conduct in the wild. Either implicitly or explicitly, conventional MMDE models predict metric scales in a discriminative manner, conditioned on the input image. With the fully supervised training scheme, they may tend to rely on the relation between training scenes and the testing scene to infer the metric scale. If the testing scene is unseen during training, it can be difficult to infer a scene scale. The MRDE task is relatively easier since it can potentially utilize more local visual clues for lower-level predictions. While establishing a global correspondence between training scenes and the testing scene might be challenging, the model can still identify local correspondences to predict relationships within different parts of a scene.

# B  More Ablation Study and Analysis

**Impact of mask prompts.** For generative painting, we randomly create mask prompts with heights $h$ equal to the input image height $H$ and widths $w \in [\alpha \cdot \min(H, W), \beta \cdot \min(H, W)]$, where $0 < \alpha < \beta < 1$ are two hyperparameters. In Tab. 7, we ablate the choices of them by painting $N = 4$ images with humans for each input. The results demonstrate that different $\alpha, \beta$ values will affect the model performance, while to a small degree. By comparing among the last three rows, we see the model performs better with smaller mask prompts within a reasonable range. When mask prompts

Table 7: Ablation study for mask sizes on the NYUv2 dataset.

| Min Ratio $\alpha$ | Max Ratio $\beta$ | $\delta_1 \uparrow$ | AbsRel $\downarrow$ | $\text{SI}_{\log} \downarrow$ | RMSE $\downarrow$ |
|---|---|---|---|---|---|
| 0.1 | 0.7 | 67.1 | 23.3 | 15.4 | 0.812 |
| 0.2 | 0.8 | 66.8 | **21.9** | 15.2 | **0.792** |
| 0.3 | 0.9 | 66.0 | 23.5 | 15.0 | 0.834 |
| 0.2 | 0.6 | 66.0 | 22.4 | **14.7** | 0.827 |
| 0.3 | 0.7 | **67.5** | 22.9 | 14.9 | 0.806 |
| 0.4 | 0.8 | 54.3 | 27.3 | 16.9 | 1.031 |

are constrained to larger sizes, as seen in the last row, model performance degrades. Moreover, when comparing rows (1st vs. 4th, 2nd vs. 5th, 3rd vs. 6th) with common centers $(\alpha + \beta)/2$ while different ranges $(\beta - \alpha)$, we observe larger ranges yields lower RMSE. This might be due to a larger range of mask sizes providing an opportunity to paint humans of more flexible sizes. As we can imagine, a close human will occupy larger areas in the painted image, and vice versa. This diversity leads to better numerical stabilities during solving the scale $s$ and the translation $t$ in Eq. (1). Therefore, we use $\alpha = 0.2, \beta = 0.8$ in our main experiments for optimal AbsRel results.

Table 8: Ablation study for different generative painting models on the NYUv2 dataset.

| Model | $\delta_1 \uparrow$ | AbsRel $\downarrow$ | $\text{SI}_{\log} \downarrow$ | RMSE $\downarrow$ |
|---|---|---|---|---|
| Stable Diffusion [57] v1.5 | 74.0 | 16.8 | 11.5 | 0.642 |
| Stable Diffusion [57] XL | 78.5 | 15.9 | 11.3 | 0.533 |
| Stable Diffusion [57] v2 | **83.2** | **13.7** | **9.78** | **0.487** |

Table 9: Ablation study for different HMR models on the NYUv2 dataset.

| Model | $\delta_1 \uparrow$ | AbsRel $\downarrow$ | $\text{SI}_{\log} \downarrow$ | RMSE $\downarrow$ |
|---|---|---|---|---|
| HMAR [62] | 82.0 | 14.2 | 9.83 | 0.489 |
| TokenHMR [63] | 80.4 | 14.9 | **9.55** | 0.495 |
| HMR 2.0 [20] | **83.2** | **13.7** | 9.78 | **0.487** |

**Impact of generative painting models.** In Tab. 8, we show the effect of using different generative painting models by generating $N = 32$ images for each input. The results indicate that current generative painting models generally work well with MfH in MMDE. MfH combined with Stable Diffusion v2 produces the best outcomes, likely due to its superior ability to generate realistic paintings. It is possible that if a generation model can better capture the real-world 2D image distributions, it has a better sense of scale, serving as a more effective source of metric scale priors. Hence, we anticipate further performance gain of MfH with more advanced generative painting models.

**Impact of HMR models.** In Tab. 9, we evaluate the impact of using different HMR models within MfH. Here we use $N = 32$ painted images for each input. Our findings indicate that the performance of our approach remains stable regardless of the HMR model used, suggesting that humans serve as effective universal landmarks for deriving metric scales from images. Furthermore, current HMR models reliably contribute to extracting metric scales for MMDE within the MfH framework.

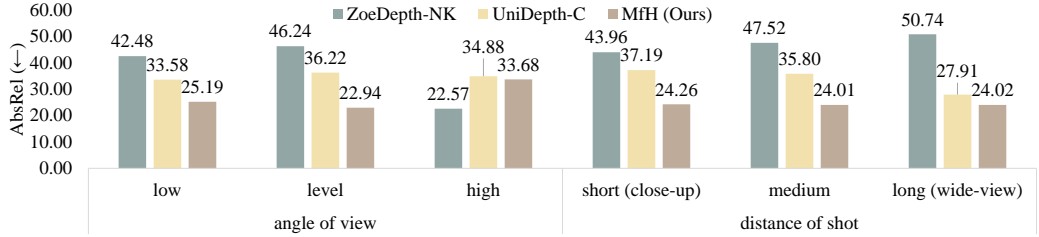

Figure 9: AbsRel ($\downarrow$) **comparisons for different types of shots on the ETH3D dataset.**

**Impact of input shot types.** To analyze the contribution of metric information from humans, we look into the MMDE results on ETH3D [54], which includes both indoor and outdoor scenes with diverse shot types. Specifically, we annotate ETH3D images with two shot-related attributes and plot the AbsRel comparisons in Fig. 9. They confirm that our MfH can robustly recover metric depths, as it consistently achieves low errors across various types of shots. We also identify that the metric information from humans helps the most for level-angle inputs. This is likely because MRDE models

tend to interpret similar semantics, such as different parts of a human body, as having similar depths. This interpretation aligns well with standing humans, which are typically generated in level-angle images. Moreover, we do not observe significant degradation with varying the distance of shots. This indicates that MfH can effectively handle general close-up and wide-view shots.

## C   Difference with Previous Methods

**Discriminative vs. Generative + Discriminative.** Traditional MMDE models metric depth prediction in a discriminative manner. That is, given an input image $I$, they model a conditional probability of metric depths $D^{\mathrm{m}}$ with a neural network $\theta$, i.e., $\mathrm{P}_\theta(D^{\mathrm{m}}|I)$. Further, the distribution of metric depths can be considered as a joint distribution of relative depths $D^{\mathrm{rel}}$ and metric scales $S$. According to the chain rule, we have

$$\mathrm{P}_\theta(D^{\mathrm{m}}|I) = \mathrm{P}_\theta(D^{\mathrm{rel}}, S|I) = \mathrm{P}_\theta(D^{\mathrm{rel}}|S, I) \cdot \mathrm{P}_\theta(S|I). \tag{9}$$

Since the relative depth is scale-invariant, $D^{\mathrm{rel}}$ is independent to $S$,

$$\mathrm{P}_\theta(D^{\mathrm{rel}}|S, I) = \mathrm{P}_\theta(D^{\mathrm{rel}}|I) \Rightarrow \mathrm{P}_\theta(D^{\mathrm{m}}|I) = \mathrm{P}_\theta(D^{\mathrm{rel}}|I) \cdot \mathrm{P}_\theta(S|I). \tag{10}$$

Most prior arts [12, 11, 14, 10, 2] parameterize the two terms $p_\theta(D^{\mathrm{rel}}|I), p_\theta(S|I)$ jointly with a single discriminative $\theta$. In contrast, we consider introducing generative and discriminative priors in modeling $\mathrm{P}(S|I)$. With the law of total probability, we have

$$\mathrm{P}(S|I) = \sum_{I^{\mathrm{paint}}} \mathrm{P}(S|I^{\mathrm{paint}}, I) \cdot \mathrm{P}(I^{\mathrm{paint}}|I), \tag{11}$$

where $I^{\mathrm{paint}}$ is the random variable for the painted image. In this equation, $\mathrm{P}(I^{\mathrm{paint}}|I)$ can be captured by the generative painting model conditioned on the input, and $\mathrm{P}(S|I^{\mathrm{paint}}, I)$ is estimated by HMR and our metric head, which are discriminative. The summation over $I^{\mathrm{paint}}$ corresponds to our global optimization upon random generative painting. Our MfH can thus approximate $\mathrm{P}(S|I)$ through Monte Carlo sampling, bridging the gap between MRDE, i.e., $\mathrm{P}(D^{\mathrm{rel}}|I)$, and MMDE, i.e., $\mathrm{P}(D^{\mathrm{m}}|I)$.

**Training vs. Pre-training + Fine-tuning vs. Test-time Adaptation** Most traditional MMDE approaches [17, 16, 15, 13, 11, 14, 10] follow a fully supervised training paradigm, for which we discuss can cause dependency to training scenes during test time. For them, expensive metric depth annotations on diverse scenes are necessary for zero-shot abilities. Some recent works seek to reduce the cost of data labeling by introducing pretraining over relative depth [12] or unlabeled image data [2]. While this mitigates the data hunger to some degree, these works also require downstream fine-tuning with metric depth annotations for MMDE. We instead consider a test-time adaptation scenario having no access to metric annotations. Under this setting, our model is required to predict the metric depth merely dependent on one input image.

## D   More Qualitative Analysis

### D.1   Case Study

Since the performance of MfH relies on the quality of the pseudo ground truths $\{\mathbf{D}_n^*\}$, we illustrate both successful and failed cases generated during this process in Fig. 10. Typical failure cases, shown in the first three rows, include 1) the generative painting model producing non-human objects with human features (1st row), 2) the generative painting model incorrectly capturing the scene scale and producing out-of-proportion humans (2nd row), and 3) the human mesh recovery model predicting meshes that penetrate each other (3rd row). In contrast, success cases in the last three rows demonstrate accurate space and scale relationships between human figures and scenes, leading to effective pseudo ground truths. These visualizations partially explain why more painted images help to improve the MMDE performance. The reason is that a larger number of painted images dilutes the influence of outliers, facilitating a more robust optimization. We hence speculate prompt engineering, as well as better sampling and filtering strategies in human painting, can further improve the performance of MfH. We leave the exploration of these to future work.

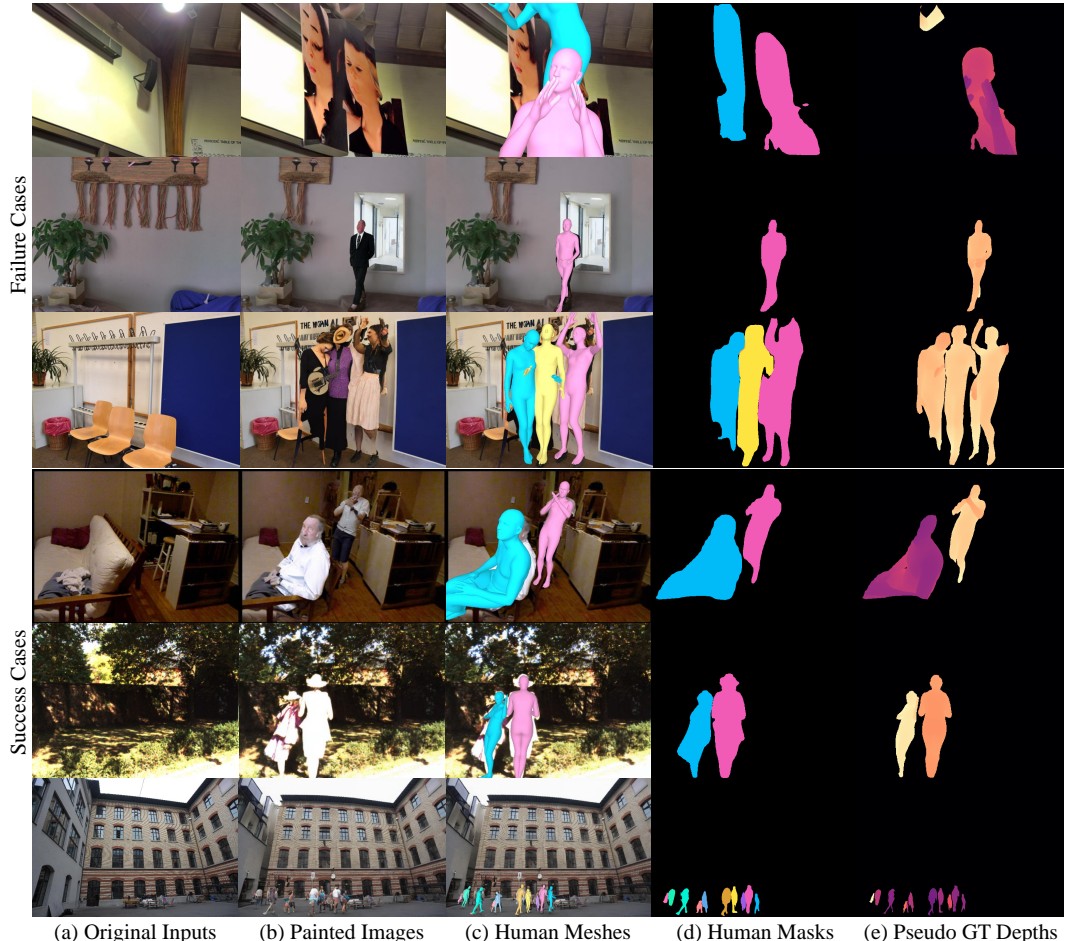

(a) Original Inputs    (b) Painted Images    (c) Human Meshes    (d) Human Masks    (e) Pseudo GT Depths

Figure 10: **Success cases and failure cases of MfH during the process of pseudo ground truth $\mathbf{D}_n^*$ generation.** The first three rows show failure cases, while the last three rows show success ones.

## D.2   User Study

We further conduct a user study for in-the-wild inputs where ground truths are unavailable. This study includes MMDE results from DepthAnything-$\{N, K\}$ [2], Metric3D-v1 [14], Metric3D-v2 [64], UniDepth-$\{C, V\}$ [10], ZeroDepth [11], ZoeDepth-NK [12], and our proposed MfH. As shown in Fig. 11, participants are presented with input images, corresponding MMDE results from the above-listed methods, along with a color bar mapping metric depth values to colors. They were instructed to select the most reasonable MMDE result for each sample, with the following guidance:

> Please choose the most reasonable metric depth estimation for each question given the input image and the meter bar. Different colors represent different metric depth values. Note that depth values farther than the maximum value or nearer than the minimum value on the meter bar are truncated.

To analyze the results, we take each input image as a separate sample and add one count to the corresponding method if its MMDE result is selected as the most reasonable MMDE given the corresponding input image and the meter bar. We then calculate the selection rate for each method, representing the proportion of selected results for this method out of the total number of selections. We received 50 responses with the results in Tab. 10. According to the meter bar attached, we roughly break down the selection rate for short, medium, and long depth ranges.

These results indicate that our MfH method achieves the highest selection rate across all depth ranges, demonstrating its robustness. Metric3D-v2 also performs well, securing the second-highest selection

Table 10: Selection rate as the most reasonable MMDE result across different ranges. The ranges indicate the maximum value of the meter bar related to each input sample.

| Range | Max Depth | DA-K [2] | DA-N [2] | M3D-v1 [14] | M3D-v2 [64] | UD-C [10] | UD-V [10] | 0D [11] | ZD-NK [12] | MfH (Ours) |
|---|---|---|---|---|---|---|---|---|---|---|
| Short-range | 10m-15m | 4.0% | 14.4% | 0.0% | 18.8% | 6.0% | 12.8% | 1.6% | 3.6% | **38.8%** |
| Medium-range | 20m-40m | 17.7% | 3.1% | 2.0% | 16.6% | 6.0% | 2.6% | 0.3% | 4.9% | **46.9%** |
| Long-range | 80m | 13.5% | 2.0% | 11.0% | 21.0% | 6.0% | 2.5% | 0.5% | 3.5% | **40.0%** |
| Overall | 10m-80m | 12.4% | 6.4% | 3.6% | 18.4% | 6.0% | 5.8% | 0.8% | 4.1% | **42.6%** |

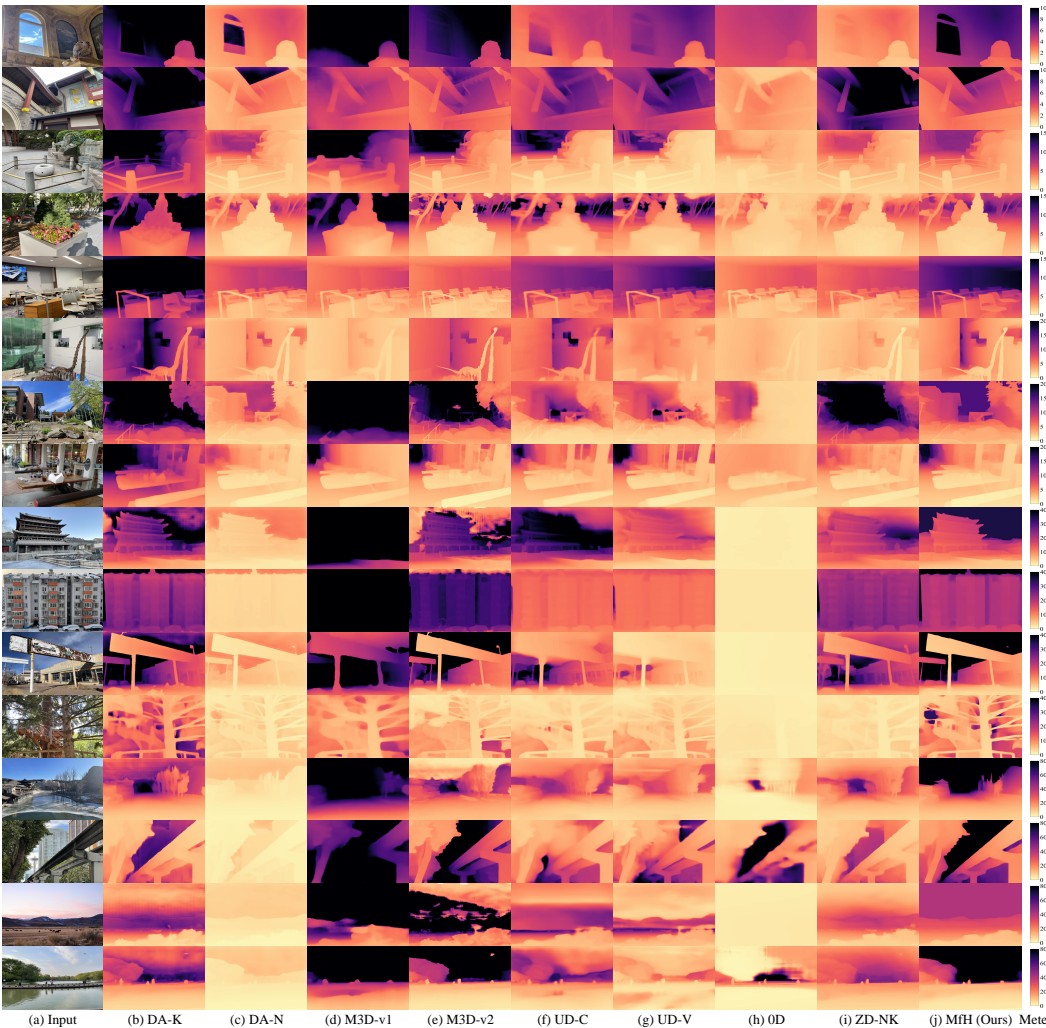

(a) Input    (b) DA-K    (c) DA-N    (d) M3D-v1    (e) M3D-v2    (f) UD-C    (g) UD-V    (h) 0D    (i) ZD-NK    (j) MfH (Ours)   Meters

Figure 11: **In-the-wild qualitative results for DSLR camera or smartphone captured images.**

rate. In contrast, other methods show variability in performance across different depth ranges. For example, DepthAnything-N has a high selection rate for short-range inputs but is not selected for inputs with larger maximum depths. This is probably due to its scene dependency. Since it is trained on NYUv2, an indoor scene dataset, its MMDE ability focuses more on short-range scenes.

## D.3 Zero-shot Qualitative Results

We demonstrate more metric depth predictions and pixel-wise AbsRel in Figs. 12 to 16.

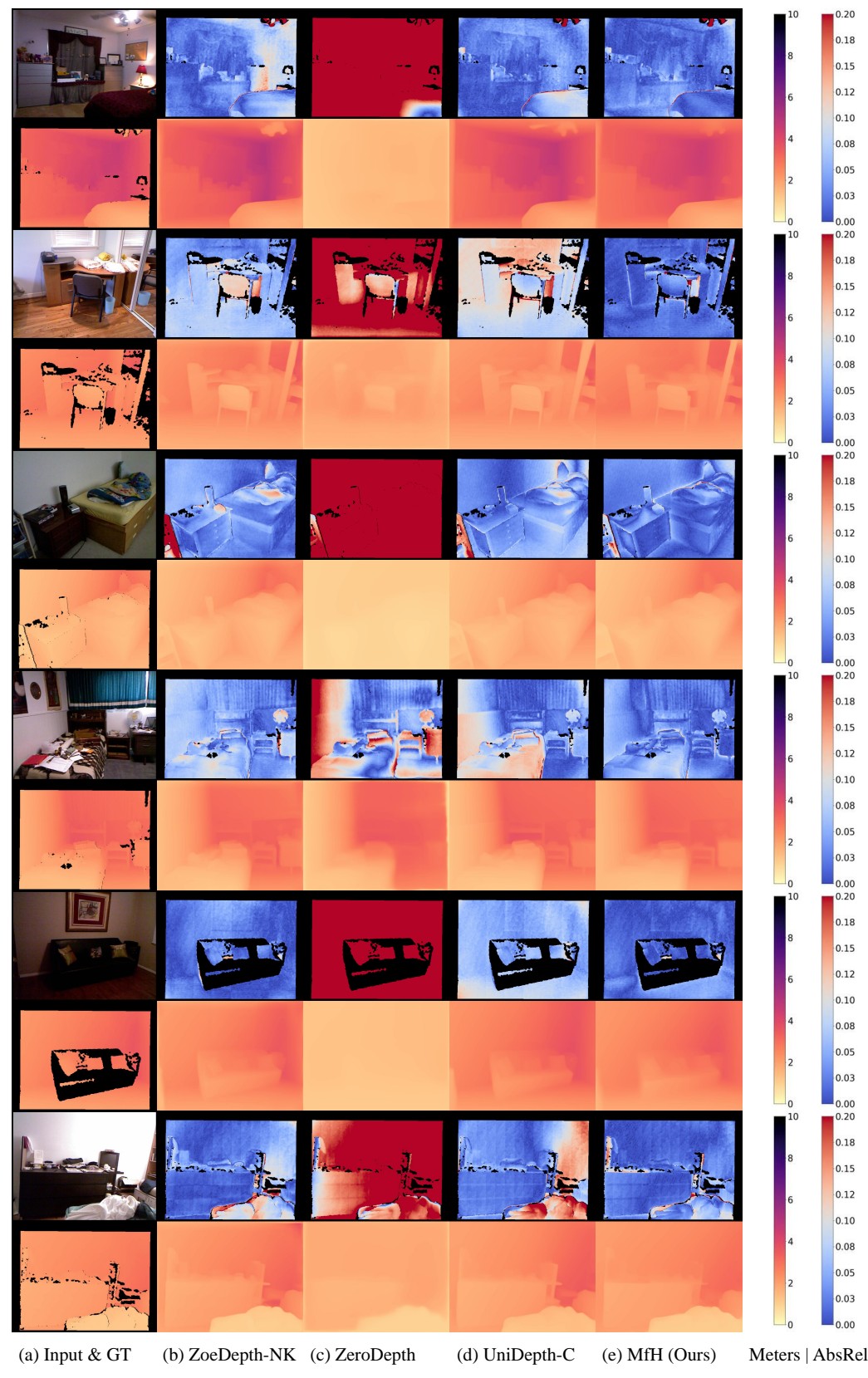

(a) Input & GT    (b) ZoeDepth-NK    (c) ZeroDepth    (d) UniDepth-C    (e) MfH (Ours)    Meters | AbsRel

Figure 12: **Zero-shot qualitative results on NYU Depth v2.**

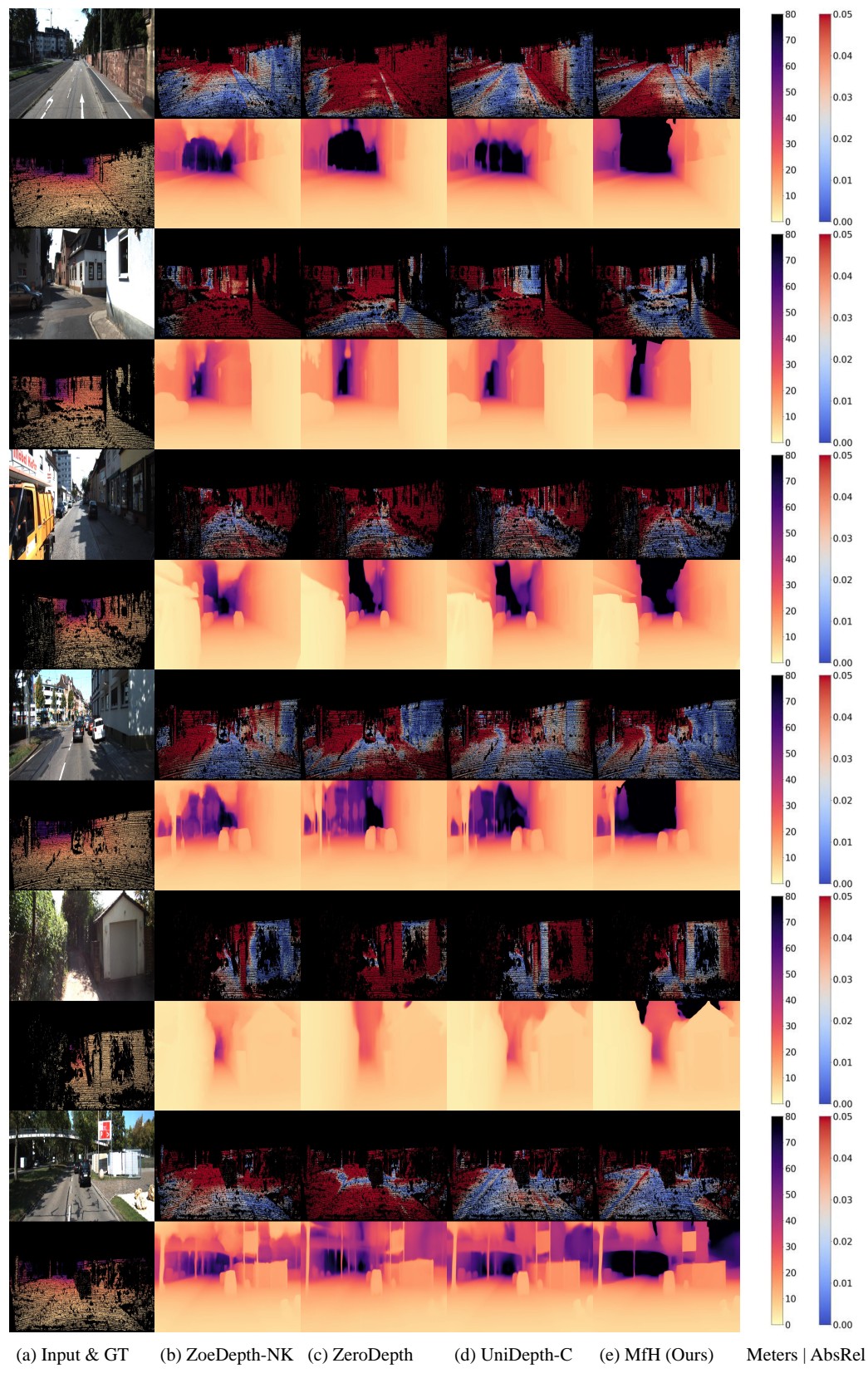

(a) Input & GT    (b) ZoeDepth-NK    (c) ZeroDepth    (d) UniDepth-C    (e) MfH (Ours)    Meters | AbsRel

Figure 13: **Zero-shot qualitative results on KITTI.**

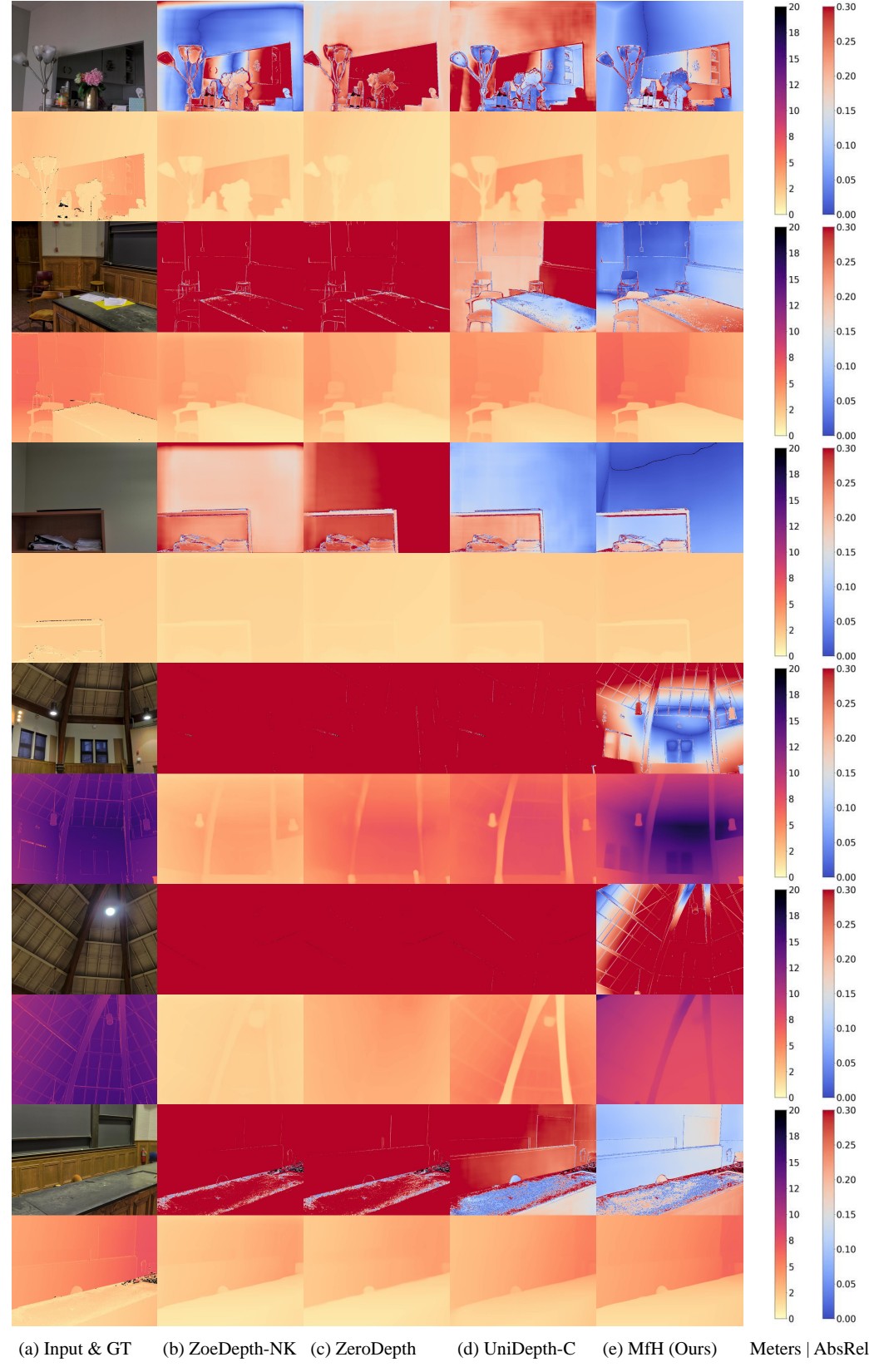

(a) Input & GT    (b) ZoeDepth-NK    (c) ZeroDepth    (d) UniDepth-C    (e) MfH (Ours)    Meters | AbsRel

Figure 14: **Zero-shot qualitative results on Diode (Indoor).**

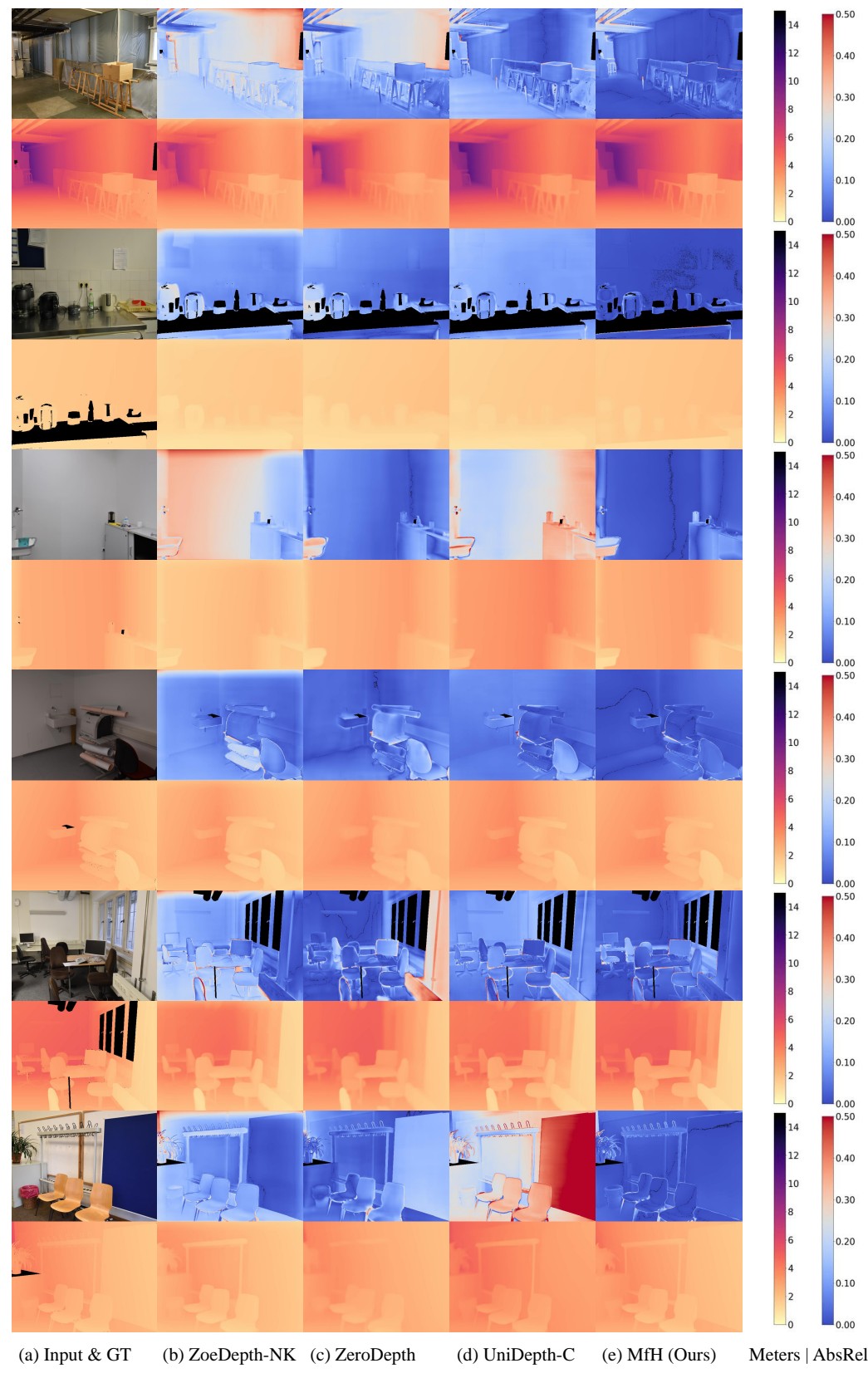

(a) Input & GT  (b) ZoeDepth-NK  (c) ZeroDepth  (d) UniDepth-C  (e) MfH (Ours)  Meters | AbsRel

Figure 15: **Zero-shot qualitative results on iBims-1.**

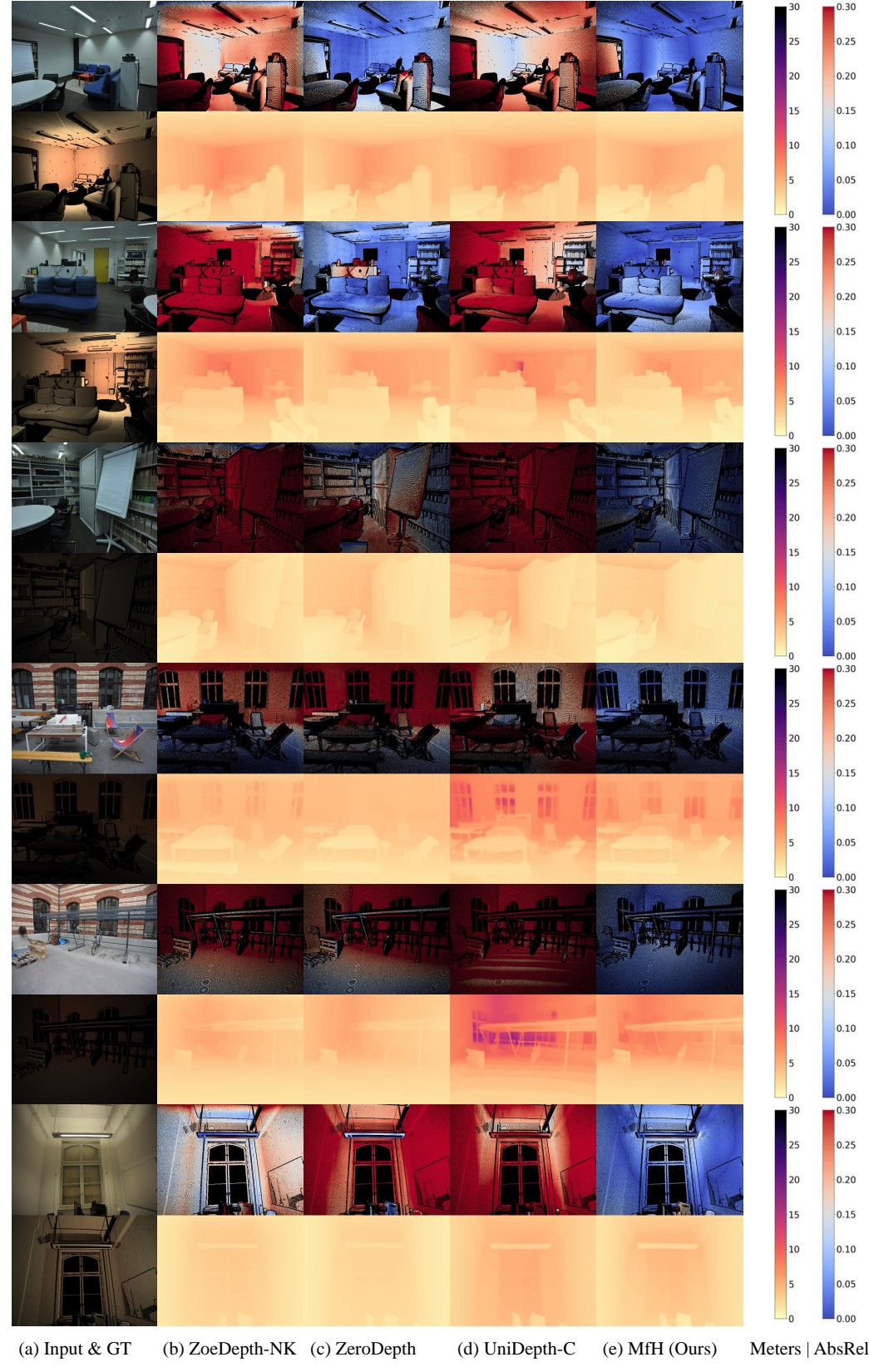

(a) Input & GT    (b) ZoeDepth-NK    (c) ZeroDepth    (d) UniDepth-C    (e) MfH (Ours)    Meters | AbsRel

Figure 16: **Zero-shot qualitative results on ETH3D.**

