# OpenReview forum: "Metric from Human: Zero-shot Monocular Metric Depth Estimation via Test-time Adaptation"
_NeurIPS.cc/2024/Conference — NeurIPS 2024 poster_

### Official Review · Reviewer_xSFg · 2024-07-01

**Soundness:** 3
**Presentation:** 3
**Contribution:** 2
**Rating:** 3
**Confidence:** 4

**Summary:**

This work proposes a test time training technique to turn a monocular relative depth estimation model into a metric monocular one. The core insight of the work is to rely on the prior of a text-to-image model (Stable Diffusion v2) to generate humans in the scene for which we are interested in knowing metric depth. Given the prior of the model, the humans should be generated in a scale aware manner and can therefore be used as hints to recover the metric scale of the scene. Using off-the-shelf methods (HMR-2) it is then possible to fit a SMPL model to the images of the humans and recover estimated metric depth for the humans. This information is finally used to train a simple linear layer that will transform a relative depth predicted by an independent model into a metric one for that specific scene. This whole process is repeated for each test image (therefore is very very slow) but does not require any metric training data and achieve decent results.

**Strengths:**

+ Original idea. While it is quite know in the literature that objects provides clues to metric monocular depth models to recover the scale of a scene (and for this reason they can be easily fooled by optical illusions) I haven’t seen before a proposal to use generative models to paint these objects (e.g., humans) when they do not occur naturally in the scene.

+ Clear motivation. I think Fig. 2 and Fig. 3 together with the introduction do a good job in explaining why the authors think that this problem is relevant and what are limitations of the existing solutions proposed in the literature.

**Weaknesses:**

1. Practicality. As reported in Fig. 5 the proposed pipeline requires up to 5 minutes per image, to ground the discussion a forward pass of the biggest depth anything model requires tens of milliseconds. Besides the latency, this work requires relying on three different models (a relative depth model, an image inpainting model and  a human mesh prediction model) to achieve metric depth prediction on a single image. I’m not convinced that these settings are realistic for any practical use case. I would suggest to the authors to explore the same idea but in an offline settings, i.e., as a way of generating pseudo metric ground truth for a dataset that does not have it.

2. Inherently limited by the support models. The proposal basically shifts the problem of recovering the scale of the scene from the depth estimation model to the image generation and human mesh estimation models. In particular the inpainting model is the one that does the heavy lifting in this work since it is tasked, provided with an arbitrary mask, to generate plausible humans. If any of the two models fail in their task the pipeline has no way of recovering as highlighted also by the authors in the failure case section of the work. If this intuition is correct, an approach that, like Marigold, starts from a pre-trained Stable diffusion and fine-tunes it for metric depth seems to be more promising with respect to the proposed solution. This solution was not explored by the authors since they didn’t want to rely on any metric depth data.

3. Works only under certain assumptions. Besides the assumption of having good behaving support model discussed in weakness (2), the proposed solution also assumes that:
   1. The definition of the area of the image to inpaint (which is a heuristic in the current implementation) picks often an area big enough and where a human can be generated. This in turns completely excludes entire categories of scenes (e.g. close ups or extremely wide views)
   2. The depth estimation models are not affected by the artifacts introduced in the scene by the inpainting methods (e.g., Fig. 6  column 5 and 6 where the inpainted images do not make much sense from a semantic point of view)

**Questions:**

a. What technique is it used for inpainting the image? (this should be specified in the paper)
b. Is Eq. 5 optimized only over the pixels with humans?

**Limitations:**

The paper discuss limitations of the current method

---

> ### Author Rebuttal · Authors · 2024-08-06
>
> Thank you for the time and effort to review our paper. Below please find our specific answers to the questions.
>
> 1. **Practicality.**
>
>     We acknowledge that MfH is not currently efficient. The runtime shown in Figure 5 is based on a sequential generate-and-estimate process, where painted images are processed one after another, illustrating a linear correlation between computational cost and the number of painted images. In Table A, we provide a breakdown of runtime with 32 images to paint when paralleled, revealing that the majority of the time is consumed by generative painting with diffusion models. Given the rapid advancements in diffusion sampling [R1, R2], we anticipate further improvements in MfH’s inference speed in the near future.
>
>     | HMR | Generative Painting | MRDE | Optimization | Total |
>     | --- | --- | --- | --- | --- |
>     | 2.4s | 5.5s | 0.1s | 0.3s | 8.3s |
>
>     Table A. Runtime breakdown for an input image.
>
>     [R1] Consistency Models, ICML 2023
>
>     [R2] One-step Diffusion with Distribution Matching Distillation, CVPR 2024
>
> 2. **Offline settings.**
>
>     We appreciate your suggestion to explore offline settings, which is a direction we find promising. Both solutions have their advantages. The offline approach requires training with abundant in-the-wild images and pseudo ground truths. While being expensive to train and prone to scene dependency, it offers fast inference. In contrast, our MfH provides a more cost-effective solution that is training-free and directly benefits from advancements in support models.
>
> 3. **Inherently limited by the support models.**
>
>     We agree that the performance of MfH is related to the support models. However, our experimental results demonstrate that MfH, using current HMR and generative painting models, can predict metric depths satisfactorily. We further conduct ablation study in Tables B and C to show the impacts of different generative painting models and HMR models. They further indicates the potential for improved MMDE results with more advanced support models.
>
>     | Model | $\delta_1$ $\uparrow$ | AbsRel $\downarrow$ | SI$_{\log}$ $\downarrow$ | RMSE $\downarrow$ |
>     | --- | --- | --- | --- | --- |
>     | SD v1.5 | 74.0 | 16.8 | 11.5 | 0.642 |
>     | SD-XL | 78.5 | 15.9 | 11.3 | 0.533 |
>     | SD v2 | 83.2 | 13.7 | 9.78 | 0.487 |
>
>     Table B. Ablation study for different generative painting models on NYUv2.
>
>     | Model | $\delta_1$ $\uparrow$ | AbsRel $\downarrow$ | SI$_{\log}$ $\downarrow$ | RMSE $\downarrow$ |
>     | --- | --- | --- | --- | --- |
>     | HMAR [R3] | 82.0 | 14.2 | 9.83 | 0.489 |
>     | TokenHMR [R4] | 80.4 | 14.9 | 9.55 | 0.495 |
>     | HMR 2.0 [20] | 83.2 | 13.7 | 9.78 | 0.487 |
>
>     Table C. Ablation study for different HMR models on NYUv2.
>
>     [R3] Tracking People by Predicting 3D Appearance, Location & Pose, CVPR 2022
>
>     [R4] TokenHMR: Advancing Human Mesh Recovery with a Tokenized Pose Representation, CVPR 2024
>
> 4. **Will failure cases lead to failure result?**
>
>     Our random generate-and-estimate process provides tolerance for failures in each component. Since the generative painting model can paint plausible humans in most cases, a few failures will not significantly impact the overall result. A sufficiently large number of painted images dilutes the influence of these failures, providing reasonable predictions. However, the idea of fine-tuning pre-trained Stable Diffusion for metric depth estimation is interesting, especially without metric depth annotations. We are excited to explore this dedicated problem in the future.
>
> 5. **Works only under certain assumptions.**
>
>     Thank you for pointing out the potential assumptions of our method. We address each concern below:
>
>     1. To assess whether MfH performs well with close-up or wide-view shots, we examine the MMDE results on the ETH3D dataset, which includes both indoor and outdoor scenes with various shot types. We annotate ETH3D images by differentiating their shot distances and plot the AbsRel comparisons in Figure R1 of the attached PDF. The results indicate no significant performance degradation when handling close-up or wide-view inputs, demonstrating that MfH can effectively handle general close-up and wide-view shots. Similar conclusion can also be drawn from the in-the-wild qualitative results in Figure R2. However, as we acknowledged in Section 5, MfH may not perform well in extreme cases where humans cannot be present in the scene.
>     2. Since MfH only aligns the MRDE of unpainted areas ($\mathbf{D}^\text{rel}$ vs. $\{\hat{\mathbf{D}}^\text{rel}_n\}$), and MMDE of human areas ($\mathbf{D}^*_n$ vs. $\{\hat{\mathbf{D}}^\text{m}_n\}$), the potential non-human artifacts are not taken into account and will not affect the optimization process. Although artifacts can be semantic meaningless, we do not observe them significantly impacting the aligned depths of other contents, as evidenced by the point clouds in Figure 6 column 5 and 6.
> 6. **Image inpainting technique.**
>
>     We adopt Stable Diffusion v2 for generative painting, which is based on Conditional Latent Diffusion [56]. Starting from a text-conditioned diffusion model checkpoint, the denoising UNet is finetuned with additional input channels for VAE-encoded masked image to result in a diffusion-based generative painting model. We will clarify this in our revised manuscript.
>
> 7. **Optimization range of Eq. 5.**
>
>     Your understanding is correct. Eq. 5 is optimized over the pixels of humans.

---

> > ### Comment · Reviewer_xSFg · 2024-08-12
> > **Checked**
> >
> > Thanks for providing a rebuttal to my criticisms.
> > I have carefully checked it and will discuss together with the other reviewers wheter to change my rating.
> >
> > I tend to agree with Reviewer 4QJ3 that the method should be directly compared with metric depth estimation methods that provide significant advantages with respect to the proposal.
> > I think the intuition behind this paper is interesting and the proposed solution original but ultimatelly there could be better ways of using these to obtain more practical models that can be executed efficiently (as pointed out in weakness 1).
> >
> > Regarding answer 5.2 my point was mroe on the fact that inpainting results will change the appearence of the scene which in turn will change the predicted monocular depth. While the MMDE loss will be optimized only on the "human" area of the image this will be in turn being modified by the global appearence of the scene. Prediction are not completelly independent per pixel. I can see how empirically this does not matter but this is another possible source of error from the proposed pipeline.

---

> > > ### Author Response · Authors · 2024-08-13
> > > **Weakness 2: More practical models that can be executed efficiently.**
> > >
> > > We acknowledge that more efficient models could potentially be developed based on our approach. However, we would like to emphasize the following points:
> > >
> > > 1. Our primary contribution is identifying the issue of scene dependency and addressing it using scene-independent metric scale priors, which have shown promising results. While there may be more efficient or effective solutions, our goal with MfH is not to present a perfect solution but to highlight a potential direction for future research.
> > > 2. Test-time adaptation differs from the traditional training-inference paradigm, where the model remains unchanged during test time. While distilling knowledge from support models might enhance efficiency during test time, it could still retain scene dependency from the training phase. In contrast, MfH adapts the MMDE model based on each test sample during optimization, reducing scene dependency and allowing for a more tailored focus on each individual test case.
> > >
> > > We appreciate your thoughtful comments, and will conduct more follow-up study based on the intuition of MfH, to make it more practical and useful in real-world applications.

---

> > > ### Author Response · Authors · 2024-08-13
> > > **Answer 5.2: Inpainting results will change the appearance of the scene.**
> > >
> > > Thank you for illustrating your point on the influence of inpainting on final results. We agree that inpainting can modify some contents of original inputs, potentially introducing noises, as you mentioned “prediction are not completely independent per pixel”. Under the framework of MfH, since we have human masks, we can use these masks to crop out human bodies and put them in the original input image. This will exclude the effect from semantic meaningless pixels.

---

> > > > ### Comment · Reviewer_xSFg · 2024-08-14
> > > > **Additional comment**
> > > >
> > > > > re: Additional Comparison with Recent Advanced MMDE Methods
> > > >
> > > > Imho a user study is not a good choice for this type of comparison, humans are notoriously bad at estimating metric depths. The nicest normalized colormap will likelly be the one that gets selected more often. A comparison with metric estimation models should be done on datasets with GT depth which is what you provided in Table R1 in the rebuttal and it does highlight the shortcoming of the proposed solution compared to trained methods.
> > > >
> > > > > re: More practical models that can be executed efficiently.
> > > >
> > > > I understand your point of view and your motivations for conducting the explorations and the advantages in terms of flexibility of test time adaptation models. My criticism are around the fact that: (a) the proposed solution is in my opinion not practical for any realistic use case and (b) relies heavily on external models for the test-time adaptation. Especially (b) from my point of view is the weakest part of the work. If Stable Diffusion fails to inpain a realistic human there's no recovery in your pipeline, so what your method does is relying heavily on the prior of the diffusion model. If this is the core contribution I go back to my original criticism that maybe a different route worth exploring would be to directly try to distill the priors of the diffusion model into a metric depth model, or use your strategy to offline label data and filter the samples where the pipeline fails.
> > > >
> > > > > re: " Under the framework of MfH, since we have human masks, we can use these masks to crop out human bodies and put them in the original input image. This will exclude the effect from semantic meaningless pixels."
> > > >
> > > > Is this what you **could do** or what you **do** in the paper?

---

> > > > > ### Author Response · Authors · 2024-08-14
> > > > >
> > > > > Thank you very much for your efforts and time in further discussing our work.
> > > > >
> > > > > > re: imho a user study is not a good choice for this type of comparison.
> > > > >
> > > > > Since Reviewer 4QJ3 pointed out the importance of robustness for in-the-wild inputs, we provide qualitative comparisons in the Figure R2 of the rebuttal PDF, and user study as a quantitative complement. The qualitative comparison demonstrates that the differences between MfH and UniDepth are not solely due to MRDE, which can be represented with normalized colormaps. Instead, people can see the difference between metric scales recovered by comparing with the meter bar. In addition, since our model uses Depth Anything as the base MRDE model, the results from MfH, Depth Anything-N, and Depth Anything-K should be similar when normalized. If, as you mentioned, "the nicest normalized colormap will likelly be the one that gets selected more often", the selection rates for them should be similar. This contradicts with what we see in the user study. In fact, the visual difference lies heavily on the metric scale recovery, for which human eyes can distinguish. The robust recovery of accurate metric scales is a key reason why MfH outperforms other models in user preference.
> > > > >
> > > > > > re:  A comparison with metric estimation models should be done on datasets with GT.
> > > > >
> > > > > Our qualitative comparisons and user study aim to demonstrate MfH’s effectiveness in real-world scenarios, which may not be fully captured by quantitative metrics alone. We hope these findings provide valuable insights for future research. As for Table R1, the comparison between MfH and many-shot methods was not intended as a direct comparison, given the varied training data, but rather to offer additional context, as suggested by Reviewer 4QJ3 "the quantitative comparison is not that important".
> > > > >
> > > > > > re: a different route worth exploring.
> > > > >
> > > > > We agree that is a route worth exploring, following the high-level idea of MfH. Meanwhile, we emphasize the importance of test-time adaptation, especially for in-the-wild usage, where it’s hard to assume that similar samples will be seen during training. Solutions like distilling priors and offline labeling do not utilize the test sample to address test-time challenges, leading to potential scene dependency like fully supervised approaches. In contrast, MfH leverages both priors from support models and the test sample to achieve more robust MMDE results, avoiding the pitfalls of the training-inference paradigm.
> > > > >
> > > > > > re: (a) the proposed solution is in my opinion not practical for any realistic use case.
> > > > >
> > > > > We consider MMDE for unseen scenes a realistic use case, where MfH leverages both priors from support models and the test sample to achieve reasonable MMDE results. This approach differs from the traditional training-inference paradigm, which may suffer from performance degradation due to scene dependency.
> > > > >
> > > > > > re: (b) relies heavily on external models for the test-time adaptation.
> > > > >
> > > > > We have acknowledged this reliance, analyzed MfH’s limitations, and proposed potential solutions and future improvements. Despite this reliance, MfH demonstrates strong performance even in in-the-wild scenarios, as evidenced by our experimental results. Moreover, this reliance offers opportunities to enhance MfH by integrating improved external models in a plug-and-play manner.
> > > > >
> > > > > > re: is this what you could do or what you do in the paper?
> > > > >
> > > > > In our paper, we did not do this. We appreciate your insight, which has inspired us to consider this solution for further improving MfH. However, this does not alter the core concept of MfH.

---

> ### Author Response · Authors · 2024-08-13
> **Additional Comparison with Recent Advanced MMDE Methods**
>
> Thank you for your valuable comments. We agree that including direct comparisons with recent advanced metric depth prediction methods would be beneficial. Since reviewer 4QJ3 emphasizes the importance of robustness on in-the-wild inputs, we also provide this comparison here as a reference.
>
> For in-the-wild inputs, where ground truths are unavailable, we further conduct a user study. This study includes all images shown in Figure R2 of the rebuttal PDF with MMDE results from UniDepth-C, UniDepth-V, ZoeDepth-NK, ZeroDepth, Metric3D-v1, Metric3D-v2, DepthAnything-N, DepthAnything-K, and our proposed MfH. Participants are presented with input images and corresponding MMDE results from all methods, along with a color bar mapping depth values to colors. They are then asked to select the most reasonable MMDE result for each input sample.
>
> To analyze the results, we take each input image as a separate sample, and add one count to the corresponding method if its MMDE result is selected as the most reasonable MMDE given the corresponding input image and the meter bar. We then calculate the selection rate for each method, representing the proportion of selected results for this method out of the total number of selections. So far, we have received 45 responses with the overall results in Table D. Further, we break down the results according to the maximum value of the meter bar as in Tables E-G.
>
> These results indicate that our MfH method achieves the highest selection rate across all depth ranges, demonstrating its robustness. Metric3D-v2 also performs well, securing the second-highest selection rate. In contrast, other methods shows variability in performance across different depth ranges. For example, DepthAnything-N has a high selection rate for short-range inputs but is not selected in inputs with larger maximum depths. This is probably due to its scene dependency. Since it is trained on NYUv2, an indoor scene dataset, its MMDE ability focus more on short-range scenes. In our revised manuscript, we will include all MMDE results (also as qualitative comparisons), these quantitative results, and discussions. We will also keep updating the results with more responses received.
>
> We hope this user study, along with Tables 1-2 in the main paper, and Table R1 and Figure R2 in the rebuttal PDF, offers a more comprehensive comparison between our MfH and recent advanced metric depth prediction methods.
>
> |  | DepthAnything-K | DepthAnything-N | Metric3D-v1 | Metric3D-v2 | UniDepth-C | UniDepth-V | ZeroDepth | ZoeDepth-NK | MfH (Ours) |
> | --- | --- | --- | --- | --- | --- | --- | --- | --- | --- |
> | Selection Rate | 12.6% | 6.3% | 3.6% | 18.2% | 6.1% | 5.4% | 0.8% | 4.3% | 42.6% |
>
> Table D. Overall selection rate as the most reasonable MMDE result.
>
> |  | DepthAnything-K | DepthAnything-N | Metric3D-v1 | Metric3D-v2 | UniDepth-C | UniDepth-V | ZeroDepth | ZoeDepth-NK | MfH (Ours) |
> | --- | --- | --- | --- | --- | --- | --- | --- | --- | --- |
> | Selection Rate | 4.0% | 13.8% | 0.0% | 18.2% | 5.3% | 12.0% | 1.8% | 3.6% | 41.3% |
>
> Table E. Selection rate as the most reasonable MMDE result for short-range (10m-15m at max) inputs.
>
> |  | DepthAnything-K | DepthAnything-N | Metric3D-v1 | Metric3D-v2 | UniDepth-C | UniDepth-V | ZeroDepth | ZoeDepth-NK | MfH (Ours) |
> | --- | --- | --- | --- | --- | --- | --- | --- | --- | --- |
> | Selection Rate | 17.8% | 3.2% | 1.9% | 16.2% | 6.7% | 2.5% | 0.3% | 5.4% | 46.0% |
>
> Table F. Selection rate as the most reasonable MMDE result for medium-range (20m-40m at max) inputs.
>
> |  | DepthAnything-K | DepthAnything-N | Metric3D-v1 | Metric3D-v2 | UniDepth-C | UniDepth-V | ZeroDepth | ZoeDepth-NK | MfH (Ours) |
> | --- | --- | --- | --- | --- | --- | --- | --- | --- | --- |
> | Selection Rate | 14.4% | 2.2% | 11.1% | 21.7% | 6.1% | 2.2% | 0.6% | 3.3% | 38.3% |
>
> Table G. Selection rate as the most reasonable MMDE result for long-range (80m at max) inputs.

---

### Official Review · Reviewer_2dzs · 2024-07-05

**Soundness:** 3
**Presentation:** 2
**Contribution:** 3
**Rating:** 6
**Confidence:** 4

**Summary:**

This paper introduces Metric-from-Human (MfH), a method to infer metric depths from images without needing metric depth annotations. Using humans as landmarks, MfH extracts scene-independent metric scale priors from generative painting models, overcoming the challenge of scene dependency in Monocular Metric Depth Estimation (MMDE). They propose a test-time adaptation framework that bridges Monocular Relative Depth Estimation (MRDE) to MMDE via a generate-and-estimate pipeline. Experiments show MfH's superior performance and generalization ability in zero-shot MMDE. The paper also addresses limitations, broader impacts, ethical considerations, and provides experimental settings and statistical significance of the results.

**Strengths:**

The paper provides a thorough analysis of the advantages and disadvantages of recently researched MMDE and MRDE models, highlighting their differences. It introduces a novel method for obtaining metric-depth in a scalable manner.

The innovative use of generative painting and Human Mesh Recovery (HMR) techniques to leverage the strong prior of human figures is a significant advantage, which has potentials to reduce the reliance on expensive metric-depth annotations.

From an architectural perspective, the protocol for incorporating metrics into MRDE appears reasonable. Additionally, the paper reasonably discusses the limitations and broader impacts of the research.

**Weaknesses:**

The experimental results presented seem to be quite poor. For example, [1] also achieves zero-shot performance on NYU or KITTI datasets, showing significantly better results than this paper. Of course, the proposed method is focused on converting MRDE to metric depth without using any metric annotations, which is a disadvantage. Therefore, I am curious whether "Metric from Human" method would also benefit MMDE models like [1].


Similarly, the metric performance seems less than ideal, potentially due to the complexity of the scenes or inaccuracies in the human prior. Therefore, a detailed and fine-grained analysis is needed to determine the accuracy of the metric information provided by humans.


##### [1] UniDepth: Universal Monocular Metric Depth Estimation

**Questions:**

Questions are included in the weakness.

**Limitations:**

As mentioned in the weaknesses, the overall metric performance is suboptimal. The method appears to be highly dependent on the performance of human recovery models and generative painting models. It is likely that in certain difficult scenes, the performance will not be maintained effectively.

---

> ### Author Rebuttal · Authors · 2024-08-06
>
> Thank you for the time and effort to review our paper. Below please find our specific answers to the questions.
>
> 1. **Experimental results.**
>
>     We acknowledge that currently our zero-shot MfH does not always outperform state-of-the-art many-shot methods. However, we would like to highlight our main contribution as pointing out the scene dependency problem of fully supervised many-shot MMDE and offering a potential solution. Our comparisons demonstrate MfH’s strong zero-shot MMDE performance across diverse scenarios, while fully supervised many-shot methods may degrade on unseen scenes. Additionally, we view MfH as a general framework for zero-shot MMDE using test-time adaptation, with performance that can be further enhanced with improved MRDE, HMR, and generative painting models.
>
> 2. **MfH can benefit MMDE models like UniDepth.**
>
>     We replace the current MRDE model with UniDepth in MfH and present the MMDE results in Tables A, B, and C below. The results show that MfH with UniDepth achieves better results on iBims-1 and ETH3D, but worse on DIODE (Indoor), than original UniDepth. This confirms strong metric scale priors extracted by MfH can enhance MMDE models on unseen scenes. The degradation on DIODE (Indoor) is because it contains extreme close-up scenes, where painting humans can meet difficulties, as we acknowledge in Section 5. For such scenarios, we see potential in incorporating objects other than humans into the generate-and-estimate pipeline as metric landmarks.
>
>     | Model | $\delta_1$ $\uparrow$ | AbsRel $\downarrow$ | SI$_{\log}$ $\downarrow$ | RMSE $\downarrow$ |
>     | --- | --- | --- | --- | --- |
>     | UniDepth-V | 79.8 | 18.1 | 10.4 | 0.760 |
>     | MfH (Ours) w/ Depth Anything | 42.2 | 34.5 | 13.2 | 1.363 |
>     | MfH (Ours) w/ UniDepth-V | 43.5 | 32.6 | 11.9 | 1.390 |
>
>     Table A. Performance comparisons of our MfH with different MRDE methods and UniDepth on DIODE (Indoor).
>
>     | Model | $\delta_1$ $\uparrow$ | AbsRel $\downarrow$ | SI$_{\log}$ $\downarrow$ | RMSE $\downarrow$ |
>     | --- | --- | --- | --- | --- |
>     | UniDepth-V | 23.4 | 35.7 | 6.87 | 1.063 |
>     | MfH (Ours) w/ Depth Anything | 67.7 | 23.3 | 9.73 | 0.738 |
>     | MfH (Ours) w/ UniDepth-V | 69.8 | 20.0 | 8.99 | 0.664 |
>
>     Table B. Performance comparisons of our MfH with different MRDE methods and UniDepth on iBims-1.
>
>     | Model | $\delta_1$ $\uparrow$ | AbsRel $\downarrow$ | SI$_{\log}$ $\downarrow$ | RMSE $\downarrow$ |
>     | --- | --- | --- | --- | --- |
>     | UniDepth-V | 27.2 | 43.1 | 8.93 | 1.950 |
>     | MfH (Ours) w/ Depth Anything | 47.1 | 24.0 | 8.16 | 1.366 |
>     | MfH (Ours) w/ UniDepth-V | 51.2 | 25.5 | 9.17 | 1.489 |
>
>     Table C. Performance comparisons of our MfH with different MRDE methods and UniDepth on ETH3D.
>
> 3. **Detailed analysis of human scale priors with respect to scenes.**
>
>     To analyze the contribution of metric information from humans, we look into the MMDE results on ETH3D, which includes both indoor and outdoor scenes with diverse type of shots. Specifically, we annotate ETH3D images with two shot-related attributes and plot the AbsRel comparisons in Figure R1 of the attached PDF. They confirm that MfH can robustly recover metric depths, as it consistently achieves low errors across various type of shots. We also identify that the metric information from humans helps the most for level-angle inputs. This is likely because MRDE models tend to interpret similar semantics, such as different parts of a human body, as having similar depths. This interpretation aligns well with standing humans, which are typically generated in level-angle images. Moreover, we do not observe significant degradation with varying the distance of shots. This indicates MfH can effectively handle general close-up and wide-view shots. We will include these analysis in our revised manuscript.
>
> 4. **Dependency on human recovery models and generative painting models.**
>
>     We acknowledge MfH depends on HMR and generative painting models. To assess their impacts, we conduct ablation study in Tables D and E. Table D demonstrates that MfH combined with stronger generative painting models yields better performance, likely due to its superior ability to generate realistic paintings. It is possible that if a generation model can better capture the real-world 2D image distributions, it has a better sense of scale, serving as a more effective source of metric scale priors. Table E indicates that different HMR models do not significantly affect MfH’s performance, probably because current HMR models can robustly assist in extracting metric information for MMDE within our MfH framework. Overall, we anticipate better MMDE results with more advanced support models, which can be integrated in a plug-and-play manner.
>
>     | Model | $\delta_1$ $\uparrow$ | AbsRel $\downarrow$ | SI$_{\log}$ $\downarrow$ | RMSE $\downarrow$ |
>     | --- | --- | --- | --- | --- |
>     | SD v1.5 | 74.0 | 16.8 | 11.5 | 0.642 |
>     | SD-XL | 78.5 | 15.9 | 11.3 | 0.533 |
>     | SD v2 | 83.2 | 13.7 | 9.78 | 0.487 |
>
>     Table D. Ablation study for different generative painting models on NYUv2.
>
>     | Model | $\delta_1$ $\uparrow$ | AbsRel $\downarrow$ | SI$_{\log}$ $\downarrow$ | RMSE $\downarrow$ |
>     | --- | --- | --- | --- | --- |
>     | HMAR [R1] | 82.0 | 14.2 | 9.83 | 0.489 |
>     | TokenHMR [R2] | 80.4 | 14.9 | 9.55 | 0.495 |
>     | HMR 2.0 [20] | 83.2 | 13.7 | 9.78 | 0.487 |
>
>     Table E. Ablation study for different HMR models on NYUv2.
>
>     [R1] Tracking People by Predicting 3D Appearance, Location & Pose, CVPR 2022
>
>     [R2] TokenHMR: Advancing Human Mesh Recovery with a Tokenized Pose Representation, CVPR 2024

---

> > ### Comment · Reviewer_2dzs · 2024-08-12
> >
> > Most of my questions have been resolved, and I sincerely appreciate the thorough responses. However, I agree with the other reviewers (4QJ3 and xSFg) that including direct comparisons with recent advanced metric depth prediction methods would be beneficial. This approach could enrich your work and provide valuable insights for future research. Overall, considering the originality of the work, I will maintain my current score.

---

> ### Author Response · Authors · 2024-08-13
> **Additional Comparison with Recent Advanced MMDE Methods**
>
> Thank you for your valuable comments and for acknowledging that most of your questions have been resolved. We agree that including direct comparisons with recent advanced metric depth prediction methods would be beneficial. Since reviewer 4QJ3 emphasizes the importance of robustness on in-the-wild inputs, we also provide this comparison here as a reference.
>
> For in-the-wild inputs, where ground truths are unavailable, we further conduct a user study. This study includes all images shown in Figure R2 of the rebuttal PDF with MMDE results from UniDepth-C, UniDepth-V, ZoeDepth-NK, ZeroDepth, Metric3D-v1, Metric3D-v2, DepthAnything-N, DepthAnything-K, and our proposed MfH. Participants are presented with input images and corresponding MMDE results from all methods, along with a color bar mapping depth values to colors. They are then asked to select the most reasonable MMDE result for each input sample.
>
> To analyze the results, we take each input image as a separate sample, and add one count to the corresponding method if its MMDE result is selected as the most reasonable MMDE given the corresponding input image and the meter bar. We then calculate the selection rate for each method, representing the proportion of selected results for this method out of the total number of selections. So far, we have received 45 responses with the overall results in Table F. Further, we break down the results according to the maximum value of the meter bar as in Tables G-I.
>
> These results indicate that our MfH method achieves the highest selection rate across all depth ranges, demonstrating its robustness. Metric3D-v2 also performs well, securing the second-highest selection rate. In contrast, other methods shows variability in performance across different depth ranges. For example, DepthAnything-N has a high selection rate for short-range inputs but is not selected in inputs with larger maximum depths. This is probably due to its scene dependency. Since it is trained on NYUv2, an indoor scene dataset, its MMDE ability focus more on short-range scenes. In our revised manuscript, we will include all MMDE results (also as qualitative comparisons), these quantitative results, and discussions. We will also keep updating the results with more responses received.
>
> We hope this user study, along with Tables 1-2 in the main paper, and Table R1 and Figure R2 in the rebuttal PDF, offers a more comprehensive comparison between our MfH and recent advanced metric depth prediction methods.
>
> |  | DepthAnything-K | DepthAnything-N | Metric3D-v1 | Metric3D-v2 | UniDepth-C | UniDepth-V | ZeroDepth | ZoeDepth-NK | MfH (Ours) |
> | --- | --- | --- | --- | --- | --- | --- | --- | --- | --- |
> | Selection Rate | 12.6% | 6.3% | 3.6% | 18.2% | 6.1% | 5.4% | 0.8% | 4.3% | 42.6% |
>
> Table F. Overall selection rate as the most reasonable MMDE result.
>
> |  | DepthAnything-K | DepthAnything-N | Metric3D-v1 | Metric3D-v2 | UniDepth-C | UniDepth-V | ZeroDepth | ZoeDepth-NK | MfH (Ours) |
> | --- | --- | --- | --- | --- | --- | --- | --- | --- | --- |
> | Selection Rate | 4.0% | 13.8% | 0.0% | 18.2% | 5.3% | 12.0% | 1.8% | 3.6% | 41.3% |
>
> Table G. Selection rate as the most reasonable MMDE result for short-range (10m-15m at max) inputs.
>
> |  | DepthAnything-K | DepthAnything-N | Metric3D-v1 | Metric3D-v2 | UniDepth-C | UniDepth-V | ZeroDepth | ZoeDepth-NK | MfH (Ours) |
> | --- | --- | --- | --- | --- | --- | --- | --- | --- | --- |
> | Selection Rate | 17.8% | 3.2% | 1.9% | 16.2% | 6.7% | 2.5% | 0.3% | 5.4% | 46.0% |
>
> Table H. Selection rate as the most reasonable MMDE result for medium-range (20m-40m at max) inputs.
>
> |  | DepthAnything-K | DepthAnything-N | Metric3D-v1 | Metric3D-v2 | UniDepth-C | UniDepth-V | ZeroDepth | ZoeDepth-NK | MfH (Ours) |
> | --- | --- | --- | --- | --- | --- | --- | --- | --- | --- |
> | Selection Rate | 14.4% | 2.2% | 11.1% | 21.7% | 6.1% | 2.2% | 0.6% | 3.3% | 38.3% |
>
> Table I. Selection rate as the most reasonable MMDE result for long-range (80m at max) inputs.

---

### Official Review · Reviewer_bPK1 · 2024-07-06

**Soundness:** 3
**Presentation:** 4
**Contribution:** 3
**Rating:** 6
**Confidence:** 4

**Summary:**

This paper enables monocular depth estimation to output metric-scale depth maps from only single images. To do this, the key idea of this paper is to leverage painting human 3D models into the input images in test-time adaptation, whose motivation is that human painter depicts subjects in consideration of scene configurations. With the painted human, the authors use it as landmarks to account for scenes in metric scale. After making an initial metric scale scene depths using the painted human, the proposed framework propagates the metric scale cues onto whole scenes using optimizations. The proposed method shows impressive performance on the zero/few-shot depth prediction over state-of-the-art methods.

**Strengths:**

The most strength point of this work is to propose a new paradigm for monocular depth estimation with metric scale. In real-world scenarios, the roughly estimated metric scale depth is enough unless precise accuracy is required. Since this work is not targeting depth estimation with precise accuracy, there is no technical issue.

In addition, the idea using both human painting and its observation is very interesting. In monocular depth estimation, to obtain metric scale, finding reasonable metric scale cue in a scene is important and not easy. This work successfully utilizes it and shows the robust performance over relevant works.

Lastly, this paper is well-written and easy to understand for readers. I hope the authors to release their source codes and pre-trained weights for the same field researchers.

**Weaknesses:**

I do not have any weakness of this paper. Please check the Questions because I have some requests for better technical descriptions.

Merely, the authors fix the references. For example, do not refer paper as Arxiv like [41] which is published in CVPR. Plus, please check the details of reference papers like [60] (0 to 0 pages).

**Questions:**

I have two requests as below:

1. Can you show me several results on images taken by the authors using DSLR and smartphone. To demonstrate the generality of this work, the results will be very helpful.

2. Following the first request, it would be interesting for the authors to do experiments on a case with an existence of radial distortions in images. Nowadays, commercial cameras, especially smartphone cameras, is not based on pinhole camera models. That means the output from Metric Head is sometimes inaccurate according to the camera used. In this consideration, the authors need to discuss the practical issue based on real-world experiments.

**Limitations:**

This paper has well described their limitations. During reading this paper, I though that these two limitations, and knew that the authors think about them.

---

> ### Author Rebuttal · Authors · 2024-08-06
>
> Thank you for the time and effort to review our paper. Below please find our specific answers to the questions.
>
> 1. **Results from DSLR and smartphone.**
>
>     We demonstrate qualitative results for DSLR and smartphone captured images in Figure R2 of the attached PDF. Depth predictions are truncated at different maximum values and displayed in color maps. These results show that our MfH can handle in-the-wild MMDE well, even for inputs with distortions, e.g., the first and last rows of the right column. Also, we observe fully supervised MMDE methods like UniDepth often provide bounded metric depths, inheriting from the limited range of sensors used in their training ground truths. In contrast, our MfH can provide more flexible results.
>
> 2. **Practical issue when the camera deviates from pinhole.**
>
>     We do observe distortions in some in-the-wild images, where using a linear metric head might not be ideal since some pixels might appear closer than the others. For general slight distortions, we expect the MRDE model, as one component of MfH, to handle these, as it leverages abundant training data which might contain distortions. To further compensate for stronger distortions, we can use a radius-related metric head to transform relative depths to metric depths, formally, the scale $s = s(r)$ and the translation $t = t(r)$. Then we believe our MfH can still work since our generate-and-estimate strategy with random painting allows local optimization of scales and translations.
>
> 3. **Reference errors.**
>
>     We appreciate your suggestion and have checked and fixed the reference errors accordingly in our revised manuscript.
>
> 4. **Source code and pre-trained weights.**
>
>     We appreciate your suggestion and will release our source code and pre-trained weights of support models upon acceptance.

---

> ### Comment · Reviewer_bPK1 · 2024-08-08
> **Checked**
>
> Thanks authors.
>
> I have checked your rebuttal, and do not have questions anymore.
>
> Best,
>
> Reviewer bPK1.

---

> > ### Author Response · Authors · 2024-08-09
> >
> > Thank you for your valuable comments. We will incorporate these extra results in our revised manuscript.

---

### Official Review · Reviewer_4QJ3 · 2024-07-08

**Soundness:** 2
**Presentation:** 3
**Contribution:** 3
**Rating:** 4
**Confidence:** 5

**Summary:**

This paper presents target zero-shot monocular metric depth estimation in the wild. They propose to use humans as landmarks to achieve metric scale and without any other information, such as focal length used in metric3D and zerodepth.  The key ideas are creative and well-motivated, addressing an important challenge in the field. While there are some limitations and areas for further exploration, the method shows clear improvements over existing approaches and opens up interesting directions for future work. However, it lacks more in-the-wild evaluation. From the comparisons, the metric accuracy is not convincing. I cannot know if the model can recover metrics in the wild.

**Strengths:**

Novelty: The paper presents an innovative approach to zero-shot monocular metric depth estimation by leveraging generative painting models and human mesh recovery. This is a creative solution to the challenge of generalizing metric depth estimation to unseen scenes.

Problem Formulation: The authors clearly articulate the limitations of current MMDE approaches, particularly their scene dependency and data hunger. The motivation for their method is well-explained and supported by empirical evidence (Fig. 2 and 3).

Method: The proposed Metric from Human (MfH) framework is well-designed and clearly explained. The use of humans as metric landmarks and the generate-and-estimate pipeline are interesting ideas.

Potential Impact: If successful, this approach could significantly advance the field of monocular metric depth estimation, enabling better generalization to unseen scenes without requiring large amounts of metric-annotated training data.

**Weaknesses:**

Lack of Detailed Results: The paper does not present any quantitative results or comparisons with existing methods in Table 1, such as Metric3D or UniDepth. This makes it difficult to assess the actual performance and advantages of the proposed approach.

Limited Discussion of Limitations: While the method is promising, there's little discussion of its potential limitations or failure cases. The method relies heavily on the performance of the generative painting and human mesh recovery models, which could introduce errors or biases.  For instance, how does it perform when the generative painting model produces unrealistic or poorly scaled humans?

Computational Complexity: Given that it involves generative painting and human mesh recovery at test time, it may be significantly slower than existing approaches. The computational cost and inference time of the test-time adaptation process are not addressed

**Questions:**

1.	You did not include comparisons with the state-of-the-art methods such as Metric3D and UniDepth in Table 1. These works include test results on the NYUv2 and KITTI datasets.
2.	You should include comparative experiments with other generative painting and human mesh recovery methods because the method relies heavily on the performance of the generative painting and human mesh recovery models, which could introduce errors or biases.
3.	Have you encountered cases where SD v2 and HMR2.0 failed to generate satisfactory human body and human mesh recovery? How did you handle situations where they couldn't produce good results?
4.	Provide more details on the generative painting model and human mesh recovery model used and how it affects the results.
5.	I noticed that in Figure 6, the human body in the Recovered Human Meshes and Painted Images point clouds do not overlap well. Please explain this phenomenon.
6.	More discussion and analysis of failure cases of the approach.

**Limitations:**

The paper lacks enough discussion of limitations. More details are provided in Questions.

---

> ### Author Rebuttal · Authors · 2024-08-06
>
> Thank you for the time and effort to review our paper. Below please find our specific answers to the questions.
>
> 1. **Comparisons with state-of-the-art methods.**
>
>     We will update Table 1 in our revised manuscript as Table R1 in the attached PDF. Our previous Table 1 aims to provide a fair comparison based on the availability of metric depth annotations (the number of shots). So we focus on zero/one/few-shot methods, excluding many-shot methods like Metric3D and UniDepth.
>
> 2. **Comparative experiments with different generative painting methods.**
>
>     In Table A, we ablate the effect of using different generative painting models in MfH. The results indicate that current generative painting models generally work well with MfH in MMDE. MfH combined with SD v2 produces the best outcomes, likely due to its superior ability to generate realistic paintings. It is possible that if a generation model can better capture the real-world 2D image distributions, it has a better sense of scale, serving as a more effective source of metric scale priors. Hence, we anticipate further performance gain of MfH with more advanced generative painting models.
>
>     | Model | $\delta_1$ $\uparrow$ | AbsRel $\downarrow$ | SI$_{\log}$ $\downarrow$ | RMSE $\downarrow$ |
>     | --- | --- | --- | --- | --- |
>     | SD v1.5 | 74.0 | 16.8 | 11.5 | 0.642 |
>     | SD-XL | 78.5 | 15.9 | 11.3 | 0.533 |
>     | SD v2 | 83.2 | 13.7 | 9.78 | 0.487 |
>
>     Table A. Comparative experiments with different generative painting models on NYUv2.
>
> 3. **Comparative experiments with different HMR methods.**
>
>     In Table B, we examine the effect of different HMR models in MfH. We find that our approach is not sensitive to such changes. This suggests that humans can serve as relatively universal landmarks for deriving metric scales from images. Also, current HMR models can robustly help extract metric scales for MMDE with our MfH framework.
>
>     | Model | $\delta_1$ $\uparrow$ | AbsRel $\downarrow$ | SI$_{\log}$ $\downarrow$ | RMSE $\downarrow$ |
>     | --- | --- | --- | --- | --- |
>     | HMAR [R1] | 82.0 | 14.2 | 9.83 | 0.489 |
>     | TokenHMR [R2] | 80.4 | 14.9 | 9.55 | 0.495 |
>     | HMR 2.0 [20] | 83.2 | 13.7 | 9.78 | 0.487 |
>
>     Table B. Comparative experiments with different HMR models on NYUv2.
>
>     [R1] Tracking People by Predicting 3D Appearance, Location & Pose, CVPR 2022
>
>     [R2] TokenHMR: Hybrid Analytical-Neural Inverse Kinematics for Whole-body Mesh Recovery
>
> 4. **Discussion and analysis of failure cases.**
>
>     We show three typical failure cases in Figure 8 of our appendix in the main PDF. They include 1) the generative painting model producing non-human objects with human-like appearances, 2) the generative painting model incorrectly capturing the scene scale and producing out-of-proportion humans, and 3) the HMR model predicting meshes that penetrate each other. Since the generative painting model can paint plausible humans in most cases, a few failures will not significantly impact the overall result. Our random generate-and-estimate process with sufficient painted images makes MfH robust to outliers. Further, we speculate prompt engineering, as well as better sampling and filtering strategies in human painting, can improve the performance of MfH.
>
> 5. **Discussion of limitations.**
>
>     We discuss the limitations of MfH in Section 5, pointing out two main assumptions MfH based on:
>
>     1. We assume humans can exist in the scene so that generative painting is possible to paint humans on the input image. While this holds for most usages of MMDE, it might not be ideal for some cases, e.g., close-up scenes. To this end, one future direction is incorporating objects other than humans into the generate-and-estimate pipeline as metric landmarks.
>     2. We assume the MRDE predictions align with true metric depths up to affine. Since the MRDE predictions can contain non-linear noises, a simple linear metric head as in MfH might not be optimal. Exploring alternative parameterizations of the metric head remains an open question.
> 6. **Computational complexity.**
>
>     We acknowledge that MfH is not currently efficient. The runtime shown in Figure 5 is based on a sequential generate-and-estimate process, where painted images are processed one after another. In Table C, we provide a breakdown of runtime with 32 images to paint when paralleled, revealing that the majority of the time is consumed by generative painting with diffusion models. Given the rapid advancements in diffusion sampling [R3, R4], we anticipate further improvements in MfH’s inference speed in the near future.
>
>     | HMR | Generative Painting | MRDE | Optimization | Total |
>     | --- | --- | --- | --- | --- |
>     | 2.4s | 5.5s | 0.1s | 0.3s | 8.3s |
>
>     Table C. Runtime breakdown for an input image.
>
>     [R3] Consistency Models, ICML 2023
>
>     [R4] One-step Diffusion with Distribution Matching Distillation, CVPR 2024
>
> 7. **Human bodies not completely overlapping with human point clouds.**
>
>     Since we only optimize a scale and a translation to convert MRDE to MMDE, each point cloud is stretched with the same set of scale and translation. Hence, we do not expect all human bodies to overlap perfectly with their corresponding point clouds but try to seek a “mode” of MRDE-to-MMDE transformation. This simplistic parameterization also provides a regularization against outliers during the generate-and-estimate process.
>
> 8. **In-the-wild evaluation.**
>
>     We present qualitative results for DSLR and smartphone captured images in Figure R2 of the attached PDF. These results demonstrate our MfH can effectively handle in-the-wild inputs. Also, we observe fully supervised MMDE methods like UniDepth often provide bounded metric depths, inheriting from the limited range of sensors used in their training ground truths. In contrast, our MfH can provide more flexible results.

---

> > ### Comment · Reviewer_4QJ3 · 2024-08-10
> > **comments**
> >
> > Thanks for authors's detailed reply. Most of my concerns have been solved.
> >
> > I still have a suggestion. As this paper aims to recover the metric depth, although with human body prior, it should compare with all recent advanced metric depth prediction methods, including unidepth, zoedepth, zerodepth, metric3d, metric3dv2, depthanything. Altough these methods may use different priors, the problem is the same. Inclusive comparisons on in-the-wild cases together can provide insight for followers. As the training data varies, the quantitative comparison is not that important. Robustness is the core problem.

---

> > > ### Author Response · Authors · 2024-08-10
> > >
> > > Thank you for your valuable feedback and for acknowledging that most of your concerns have been addressed.
> > >
> > > We agree that in-the-wild comparisons can offer significant insights for our readers. While the 1-page limit of the attached PDF (with no anonymous link allowed) only permitted us to present partial results in Figure R2, where we demonstrate the robustness of our approach, we will ensure to include more comprehensive in-the-wild comparisons with all recent advanced metric depth prediction methods as you mentioned in the revised manuscript.

---

> ### Author Response · Authors · 2024-08-13
> **Additional Comparison with Recent Advanced MMDE Methods**
>
> For in-the-wild inputs, where ground truths are unavailable, we further conduct a user study. This study includes all images shown in Figure R2 of the rebuttal PDF with MMDE results from UniDepth-C, UniDepth-V, ZoeDepth-NK, ZeroDepth, Metric3D-v1, Metric3D-v2, DepthAnything-N, DepthAnything-K, and our proposed MfH. Participants are presented with input images and corresponding MMDE results from all methods, along with a color bar mapping depth values to colors. They are then asked to select the most reasonable MMDE result for each input sample.
>
> To analyze the results, we take each input image as a separate sample, and add one count to the corresponding method if its MMDE result is selected as the most reasonable MMDE given the corresponding input image and the meter bar. We then calculate the selection rate for each method, representing the proportion of selected results for this method out of the total number of selections. So far, we have received 45 responses with the overall results in Table D. Further, we break down the results according to the maximum value of the meter bar as in Tables E-G.
>
> These results indicate that our MfH method achieves the highest selection rate across all depth ranges, demonstrating its robustness. Metric3D-v2 also performs well, securing the second-highest selection rate. In contrast, other methods shows variability in performance across different depth ranges. For example, DepthAnything-N has a high selection rate for short-range inputs but is not selected in inputs with larger maximum depths. This is probably due to its scene dependency. Since it is trained on NYUv2, an indoor scene dataset, its MMDE ability focus more on short-range scenes. In our revised manuscript, we will include all MMDE results (also as qualitative comparisons), these quantitative results, and discussions. We will also keep updating the results with more responses received.
>
> We hope this user study, along with Tables 1-2 in the main paper, and Table R1 and Figure R2 in the rebuttal PDF, offers a more comprehensive comparison between our MfH and recent advanced metric depth prediction methods. We sincerely hope this addresses your concerns regarding in-the-wild comparisons, and will appreciate it if you could kindly reconsider the rating. Thank you.
>
> |  | DepthAnything-K | DepthAnything-N | Metric3D-v1 | Metric3D-v2 | UniDepth-C | UniDepth-V | ZeroDepth | ZoeDepth-NK | MfH (Ours) |
> | --- | --- | --- | --- | --- | --- | --- | --- | --- | --- |
> | Selection Rate | 12.6% | 6.3% | 3.6% | 18.2% | 6.1% | 5.4% | 0.8% | 4.3% | 42.6% |
>
> Table D. Overall selection rate as the most reasonable MMDE result.
>
> |  | DepthAnything-K | DepthAnything-N | Metric3D-v1 | Metric3D-v2 | UniDepth-C | UniDepth-V | ZeroDepth | ZoeDepth-NK | MfH (Ours) |
> | --- | --- | --- | --- | --- | --- | --- | --- | --- | --- |
> | Selection Rate | 4.0% | 13.8% | 0.0% | 18.2% | 5.3% | 12.0% | 1.8% | 3.6% | 41.3% |
>
> Table E. Selection rate as the most reasonable MMDE result for short-range (10m-15m at max) inputs.
>
> |  | DepthAnything-K | DepthAnything-N | Metric3D-v1 | Metric3D-v2 | UniDepth-C | UniDepth-V | ZeroDepth | ZoeDepth-NK | MfH (Ours) |
> | --- | --- | --- | --- | --- | --- | --- | --- | --- | --- |
> | Selection Rate | 17.8% | 3.2% | 1.9% | 16.2% | 6.7% | 2.5% | 0.3% | 5.4% | 46.0% |
>
> Table F. Selection rate as the most reasonable MMDE result for medium-range (20m-40m at max) inputs.
>
> |  | DepthAnything-K | DepthAnything-N | Metric3D-v1 | Metric3D-v2 | UniDepth-C | UniDepth-V | ZeroDepth | ZoeDepth-NK | MfH (Ours) |
> | --- | --- | --- | --- | --- | --- | --- | --- | --- | --- |
> | Selection Rate | 14.4% | 2.2% | 11.1% | 21.7% | 6.1% | 2.2% | 0.6% | 3.3% | 38.3% |
>
> Table G. Selection rate as the most reasonable MMDE result for long-range (80m at max) inputs.

---

### Author Rebuttal · Authors · 2024-08-06

We sincerely thank all reviewers for their insightful feedback. They acknowledge the results as showing clear improvements (4QJ3), impressive (bPK1), superior (2dzs), and decent (xSFg). Reviewer 4QJ3 further finds our key idea creative and well-motivated while reviewer bPK1 sees our method establishing a new paradigm.

We include Table R1 and Figures R1 and R2 in the PDF file attached, showing more complete comparisons with state-of-the-art models, performance comparisons with respect to different types of shots, and more in-the-wild qualitative results respectively. Below we separately address concerns raised in the reviews. We hope our responses could clarify your confusion, and are more than happy to provide further explanations if needed.

---

### Author Response · Authors · 2024-08-14

Dear Reviewers,

Thank you once again for your thorough review and thoughtful feedback on our paper. As the author-reviewer discussion period is coming to a close, we were wondering if you have any further questions or concerns based on our previous response. Please feel free to let us know with any additional comments.

Thank you,

The Authors

---

### Decision · Program_Chairs · 2024-09-25

**Decision:**

Accept (poster)

**Comment:**

The paper explores interesting observations that generative models can usually hallucinate humans suitable for a context, both indoor and outdoor, and we have strong structural priors for the shapes of human bodies. Those two observations can potentially lead to new ways of monocular metric depth estimation. This paper explores the insights by utilizing the external generative models and human 3D pose models. The results are quite promising. Although Reviewer xSFg clearly pointed out valid concerns of using the proposed approach in practical scenarios, the AC believes that this exploration can be interesting for the machine learning community and potentially inspire future works to generate practical methods with the ideas.